# Clinical trial links oncolytic immunoactivation to survival in glioblastoma

Alexander L. Ling[1,17], Isaac H. Solomon[2,3,17], Ana Montalvo Landivar[1,17], Hiroshi Nakashima[1,17], Jared K. Woods[2,3], Andres Santos[1,2,3], Nafisa Masud[1], Geoffrey Fell[4], Xiaokui Mo[5,6], Ayse S. Yilmaz[5,6], James Grant[1], Abigail Zhang[1], Joshua D. Bernstock[1], Erickson Torio[1], Hirotaka Ito[1], Junfeng Liu[1], Naoyuki Shono[1], Michal O. Nowicki[1], Daniel Triggs[1], Patrick Halloran[1], Raziye Piranlioglu[1], Himanshu Soni[1], Brittany Stopa[1], Wenya Linda Bi[1], Pierpaolo Peruzzi[1], Ethan Chen[1], Seth W. Malinowski[2,3], Michael C. Prabhu[2,3], Yu Zeng[2,3], Anne Carlisle[7], Scott J. Rodig[2,3,7], Patrick Y. Wen[8], Eudocia Quant Lee[8], Lakshmi Nayak[8], Ugonma Chukwueke[8], L. Nicolas Gonzalez Castro[8,9], Sydney D. Dumont[10,11], Tracy Batchelor[9], Kara Kittelberger[12], Ekaterina Tikhonova[13], Natalia Miheecheva[13], Dmitry Tabakov[13], Nara Shin[13], Alisa Gorbacheva[13], Artemy Shumskiy[13], Felix Frenkel[13], Estuardo Aguilar-Cordova[14], Laura K. Aguilar[14], David Krisky[14], James Wechuck[14], Andrea Manzanera[14], Chris Matheny[14], Paul P. Tak[14], Francesca Barone[14], Daniel Kovarsky[15], Itay Tirosh[15], Mario L. Suvà[10,11], Kai W. Wucherpfennig[16], Keith Ligon[2,3], David A. Reardon[8] & E. Antonio Chiocca[1✉]

Immunotherapy failures can result from the highly suppressive tumour microenvironment that characterizes aggressive forms of cancer such as recurrent glioblastoma (rGBM)[1,2]. Here we report the results of a first-in-human phase I trial in 41 patients with rGBM who were injected with CAN-3110—an oncolytic herpes virus (oHSV)[3]. In contrast to other clinical oHSVs, CAN-3110 retains the viral neurovirulence *ICP34.5* gene transcribed by a nestin promoter; nestin is overexpressed in GBM and other invasive tumours, but not in the adult brain or healthy differentiated tissue[4]. These modifications confer CAN-3110 with preferential tumour replication. No dose-limiting toxicities were encountered. Positive HSV1 serology was significantly associated with both improved survival and clearance of CAN-3110 from injected tumours. Survival after treatment, particularly in individuals seropositive for HSV1, was significantly associated with (1) changes in tumour/PBMC T cell counts and clonal diversity, (2) peripheral expansion/contraction of specific T cell clonotypes; and (3) tumour transcriptomic signatures of immune activation. These results provide human validation that intralesional oHSV treatment enhances anticancer immune responses even in immunosuppressive tumour microenvironments, particularly in individuals with cognate serology to the injected virus. This provides a biological rationale for use of this oncolytic modality in cancers that are otherwise unresponsive to immunotherapy (ClinicalTrials.gov: NCT03152318).

High-grade gliomas (HGGs) are central nervous system tumours of glial origin with highly malignant morphologic and genetic features[5,6]. Among these, GBM is characterized by the worst outcome in terms of survival, with rapid recurrence after neurosurgical resection and chemoradiation[7]. Recurrent HGG (rHGG), including recurrent GBM (rGBM), is characterized by rapid neurological morbidity and survival of less than 10 months[8]. Although much is known of the genetics, cellular composition and evolution of HGG/GBM, this has not translated into successful therapies. Traditional immunotherapy has also been ineffective in rHGG/rGBM[1]. This is thought to be due to the scarcity of infiltrating antitumour lymphocytes caused by a highly immunosuppressive tumour microenvironment (TME), defining these tumours as 'lymphocyte depleted'[2]. For rGBMs and several other highly immunosuppressive solid cancers, there is a need to find treatment modalities that can convert the TME into one that is more amenable to immunotherapy and immune activation.

[1]Harvey Cushing Neuro-oncology Laboratories, Department of Neurosurgery, Brigham and Women's Hospital, Boston, MA, USA. [2]Department of Pathology, Brigham and Women's Hospital, Boston, MA, USA. [3]Department of Pathology, Dana-Farber Cancer Institute, Boston, MA, USA. [4]Department of Biostatistics, Dana-Farber Cancer Institute, Boston, MA, USA. [5]Center for Biostatistics, Department of Biomedical Informatics, The Ohio State University, Columbus, OH, USA. [6]James Comprehensive Cancer Center, The Ohio State University, Columbus, OH, USA. [7]Center for Immuno-Oncology, Dana-Farber Cancer Institute, Boston, MA, USA. [8]Center for Neuro-oncology, Dana-Farber Cancer Institute, Boston, MA, USA. [9]Department of Neurology, Brigham and Women's Hospital, Boston, MA, USA. [10]Department of Pathology and Center for Cancer Research, Massachusetts General Hospital, Boston, MA, USA. [11]Broad Institute of MIT and Harvard, Cambridge, MA, USA. [12]ClearPoint Neuro, Solana Beach, CA, USA. [13]BostonGene, Waltham, MA, USA. [14]Candel Therapeutics, Needham, MA, USA. [15]Department of Molecular Cell Biology, Weizmann Institute of Medical Sciences, Tel Aviv, Israel. [16]Department of Cancer Immunology and Virology, Dana Farber Cancer Institute, Boston, MA, USA. [17]These authors contributed equally: Alexander L. Ling, Isaac H. Solomon, Ana Montalvo Landivar, Hiroshi Nakashima. ✉e-mail: eachiocca@bwh.harvard.edu

Oncolytic viruses are a form of immunotherapy in which oncolytic-virus-induced oncolysis alters the TME, promoting proinflammatory pathways, activating resident and newly recruited immune cells through exposure of viral and possibly tumour antigens[9–13]. Several oncolytic viruses have been and continue to be tested in oncology, with one approved as a single-agent intralesional injection into melanoma[14] and a second one approved for injection into rGBM in Japan[15–17]. Notably, several early-phase oncolytic-virus clinical trials for HGG have been published in recent high-profile literature[17–23]. Yet, immunological profiling of rGBMs treated with oncolytic viruses in numbers sufficient to correlate with a therapeutic outcome has been lacking.

Here we report safety data for a first-in-human phase I clinical trial in 41 patients with rHGG/rGBM who were treated with CAN-3110—an oncolytic virus derived from herpes simplex virus type 1 (oncolytic HSV (oHSV); ClinicalTrials.gov: NCT03152318). We found that patients whose survival response after CAN-3110 was the longest were characterized by positive HSV1 serology with CAN-3110 clearance from infected tumour, differences in T cell clonotype metrics, and tumour transcriptomic signatures associated with immune activation programs. These findings provide human immunological and biological evidence supporting intralesional oncolytic treatment modalities to change the immunosuppressive TME into one that is more favourable for immunotherapy, providing broad relevance for the therapy of many solid cancers that are otherwise impervious to immune rejection.

## Safety of CAN-3110 in patients with rHGG/rGBM

Most clinical oHSVs to date have deleted or removed the viral gene encoding ICP34.5 (refs. 3,4); although ICP34.5 enables robust replication of HSV in infected cells[24,25], it is also responsible for neurotoxicity in mice[26]. To take advantage of ICP34.5's functions that enhance viral replication/persistence and minimize neurotoxicity, CAN-3110 (former designation, rQNestin34.5v.2) was engineered to express a copy of the viral *ICP34.5* gene under transcriptional control of the promoter for nestin, restricting viral replication and virulence to HGG/GBM cells[3,4]. To further ensure safety for initial use in humans, a multi-cohort clinical trial design was implemented (Extended Data Fig. 1a). Moreover, to ensure that the injections occurred in tumour, intraoperative MRI guidance was used to visualize the injections (Extended Data Fig. 1b,c and Supplementary Methods). A total of 41 patients with rHGG/rGBM (42 interventions, see the note on participant 042/054 in the Supplementary Methods; Extended Data Tables 1 and 2) were recruited to the trial. The patients were enrolled at their first ($n = 18$), second ($n = 9$) or third ($n = 3$) recurrence for cohorts 1–9 and at the first ($n = 5$), second ($n = 3$), third ($n = 1$) or fourth ($n = 3$) recurrence for cohort 10 (Extended Data Table 3). Tumour genomic data were typical for a rHGG/rGBM population (Extended Data Fig. 2), including the presence of mutations in the *CDKN2A/B* (encoding p16) tumour suppressor pathway, previously shown to complement viral replication of oHSVs, such as CAN-3110, with defects in the viral ribonucleotide reductase function[27]. Serious adverse events, consisting of seizures requiring hospitalization and intervention, were observed in two patients, but there were no dose-limiting toxicities or clinical/pathological evidence of ICP34.5-induced HSV1 encephalitis/meningitis (Tables 1 and 2 and Extended Data Table 4). Thus, these data indicate the relative human safety of CAN-3110 at all tested doses despite the presence of the HSV1 *ICP34.5* neurovirulence gene.

## HSV1 serology predicts efficacy

We tried to determine whether there were patients who benefited the most from treatment. Notably, 9 out of 41 patients (22%) had tumours associated with reduced survival[28–30], such as depth (insular, thalamic),

**Table 1 | Total adverse events (grade 1 or 2) related to CAN-3110**

| Category | CTC grade 1 | CTC grade 2 |
|---|---|---|
| Blood and lymphatic systems disorders | | |
| Low eosinophil count | 1 | 0 |
| General disorders and administration site conditions | | |
| Fatigue | 1 | 0 |
| Fever | 3 | 0 |
| Investigations | | |
| Alanine aminotransferase increased | 1 | 0 |
| Lymphocyte count decreased | 1 | 0 |
| Platelet count decreased | 1 | 0 |
| Musculoskeletal and connective tissue disorders | | |
| Muscle weakness—lower limb | 1 | 0 |
| Muscle weakness—upper limb | 1 | 0 |
| Nervous systems disorders | | |
| Cerebral oedema | 2 | 1 |
| Headache | 0 | 1 |
| Expressive aphasia | 1 | 0 |
| Left leg numbness | 0 | 1 |
| Left visual field defect | 0 | 1 |
| Right arm joint position sense loss | 1 | 0 |
| Seizure | 0 | 1 |
| Speech | 0 | 1 |

Events reported as of 18 April 2022.

multifocality/multicentricity or bilateral laterality. In these latter cases, only one of the tumours or one hemispheric side of tumour was injected. Notably, patients like these are not routinely eligible for clinical trials, compounding the difficulty in comparing to historical clinical trial data. The estimated median overall survival (mOS) of the entire rHGG/rGBM group was 11.6 months (95% confidence interval (CI) = 7.8–14.9 months) (Fig. 1a). On the basis of the latest WHO classification[5], we observed that, for the isocitrate dehydrogenase (*IDH1/2*) wild-type (WT) rGBM subgroup ($n = 32$ patients, 33 interventions), the mOS was 10.9 months (95% CI = 6.9–14.4 months), whereas, for the subgroup with recurrent *IDH* mutant (*IDH^mut*) anaplastic astrocytoma (rAA; grade 3 or 4) ($n = 4$), the mOS was 5.4 months (95% CI = 2.6–∞ months) and, for the recurrent anaplastic oligodendroglioma (*IDH^mut*; 1p/19q co-deleted), the mOS was 39.9 months (95% CI = 39.9–∞ months) ($n = 5$) (Fig. 1b). Progression-free survival times for the entire cohort and the cohort divided by the three rHGG diagnostic groups are shown in Extended Data Fig. 3a,b, respectively, and the clinical course of treated patients is shown in Extended Data Fig. 3c,d. Note that, in the swimmer plots, the timepoint of post-injection tumour resection is illustrated by a coloured triangle, with most additional antitumour therapies administered after resection. Full patient treatment histories have been included in Supplementary Table 1. Examples of significant clinical and radiographic responses are illustrated in Extended Data Fig. 4, including a response in a multifocal/multicentric rGBM.

Clinical trials of oncolytic-virus therapy in cancer have not shown that viral serology predicts response[19,31]. We checked whether HSV1 serology or seroconversion predicted survival in our study. In total, 14 out of 41 patients were seronegative for HSV1 before CAN-3110 treatment, with 4 out of 14 patients seroconverting after (Extended Data Table 3). Given the impact of *IDH^mut* on survival[32] and the small number of *IDH^mut* patients in the study, we focused analyses on the patients with *IDH^WT* rGBM. Notably, HSV1 seropositivity both before and after treatment was associated with significantly longer survival after treatment ($P = 0.009$ and $P = 0.007$, respectively) (Extended Data Fig. 5a).

**Table 2 | Serious adverse events (grade 3 or above) possibly, likely or definitely related to CAN-3110**

| Case | Dose cohort | Days after CAN-3110 | Category | Adverse event | CTC grade | Relation to CAN-3110 | SUSAR |
|------|-------------|---------------------|----------|---------------|-----------|----------------------|-------|
| 033 | Arm A 3×10⁹ | 16 | Nervous system disorders | Seizure | 3 | Possible | N |
| 033 | Arm A 3×10⁹ | 21 | Nervous system disorders | Cerebral haematoma | 3 | Possible | N |
| 046 | Arm A 1×10⁹ (2 ml) | 2 | Nervous system disorders | Seizure | 3 | Possible | N |
| 046 | Arm A 1×10⁹ (2 ml) | 3 | Nervous system disorders | Muscle weakness, left-sided | 3 | Possible | N |
| | | | Nervous system disorders | Muscle weakness, facial muscle | 3 | Possible | N |

SUSAR, suspected unexpected serious adverse reaction.

In a survival analysis, HSV1-seropositive patients lived a median of 14.2 months (95% CI = 9.5–15.7 months) versus only 7.8 months (95% CI = 3.0–∞ months) for seronegative patients ($P = 0.007$, likelihood ratio test; Fig. 1c). By contrast, HSV2 serology was not associated with survival ($P = 0.9$, likelihood ratio test; Fig. 1d). Similarly, the trend towards longer survival for HSV1-seropositive patients was observed in the small number of patients with $IDH^{mut}$rAA (Extended Data Fig. 5b). Cox proportional hazard analyses in $IDH^{WT}$ rGBMs validated pre-CAN-3110 positive HSV1 serology as a highly significant independent predictor of survival (Fig. 1e). As previously reported, age and tumour volume were also independent survival predictors[33,34]. These results therefore suggest the importance of an immunological mechanism for the response of patients with $IDH^{WT}$ rGBM to CAN-3110 therapy.

## CAN-3110 increases T cells in tumours

There has been understandable reluctance to routinely collect rHGGs/rGBMs after an experimental therapy as it requires a surgical procedure. Even post-mortem examinations are rarely performed. To determine whether CAN-3110 induced a significant increase in lymphocytes in this lymphocyte-depleted tumour[2], we endeavoured to recover as many post-treatment tumours as feasible either by re-resections at suspected progression and/or by post-mortem. Paired tumours from before and various timepoints after CAN-3110 treatment were analysed for a majority of separate rHGGs/rGBMs from patients after CAN-3110 treatment (Supplementary Table 2a–c and Supplementary Methods). In total, all analysed (except one) tumour pairs retained immunohistochemical expression for nestin and nectin-1, one of the major HSV receptors in cells[35], both before and after injection (one tumour pair had insufficient material for pre-injection immunohistochemistry analysis) (Extended Data Fig. 6a,b and Supplementary Table 2b). Histological and immunohistochemical analyses showed increases in CD8⁺ and CD4⁺ tumour-infiltrating lymphocytes (TILs) in most paired tumours after CAN-3110 treatment (Extended Data Fig. 6c and Supplementary Table 2b). TILs could be visualized in a perivascular distribution, as well as with diffusely scattered cells and occasional clusters throughout the tumour (Extended Data Fig. 6d) and surrounding large areas of tumour necrosis (Extended Data Fig. 6e). Quantitative analyses showed a significant increase in CD4⁺ ($P = 0.00085$) and in CD8⁺ ($P = 0.0034$) TILs in most analysed paired tumours after CAN-3110 treatment (Fig. 2a and Supplementary Table 2c). There was a non-significant trend in CD20⁺ B cell increases in almost half of post-treatment samples. The most significant increases in CD8⁺ and CD4⁺ T cells were adjacent to perinecrotic areas that were possibly due to CAN-3110 cytotoxicity (Fig. 2b). The observed post-treatment increases in CD8⁺ and CD4⁺ T cells were significantly correlated with post-treatment survival in $IDH^{WT}$ rGBMs, but only in HSV1-seropositive patients ($r = 0.58$, $P = 0.017$ (CD8⁺) and $r = 0.57$, $P = 0.026$ (CD4⁺); Fig. 2c). Importantly, the overall quantitative assessments of CD8⁺, CD4⁺ and CD20⁺ TILs used in this analysis were not significantly confounded by

the time of tissue collection (Extended Data Fig. 7a–c). Furthermore, longitudinal analyses of patient immune counts over time showed a non-significant trend towards a time-dependent decrease in CD8⁺ T cell numbers (albeit, without much change in CD4⁺ or B cells) over several months in HSV1-seronegative patients (Kruskal–Wallis test, $P = 0.16$; Extended Data Fig. 7d,e) more so than in HSV1-seropositive patients ($P = 0.45$), suggesting that the immune response induced by CAN-3110 may be durable over long periods of time in the latter. Multiplex immunofluorescence analysis in two of the analysed patients also showed CD68⁺ macrophage populations (specifically CD68⁺CD163⁺ myeloid cells expressing PD-L1) after CAN-3110 treatment, particularly in perinecrotic tumour regions (Extended Data Fig. 7f–i). These results therefore indicate that CAN-3110 induced an increase in TILs that was associated with longer survival in HSV1-seropositive patients but not in HSV1-seronegative patients.

## Persistence is linked to seronegativity

It has been rare to find oncolytic viruses in injected tumours and, even when observed, persistence is limited to a few weeks[21]. We examined whether the observed immune infiltrates were associated with oHSV persistence in injected tumours. In 12 out of 29 tumours, oHSV antigen was present even several months after CAN-3110 injection (with the longest at 801 days) (Fig. 3a and Supplementary Table 2c). Importantly, in one case of multicentric GBM, a non-injected temporal lesion analysed 8 months after CAN-3110 injection showed positivity for HSV antigen in the absence of antigen detection in the original injected lesion (Fig. 3b). PCR was used to confirm the presence of CAN-3110-specific viral DNA, indicating probable ongoing replication, and spread from the injected lesion to the non-injected tumour (Extended Data Fig. 8). Coupled with the previous findings, these results showed that there was prolonged persistence of CAN-3110 in some patients, with increased CD4⁺ and CD8⁺ T cells in injected rHGGs in most participants and evidence of ongoing replication even in a tumour that was not initially injected in a patient with multicentric rGBM.

We examined whether the prolonged persistence of CAN-3110 in injected tumours was associated with HSV1 serological status. Indeed, oHSV persistence was significantly correlated with the absence of HSV1 seropositivity either before or after CAN-3110 treatment (Fig. 3c,d). These findings suggested that oHSV persistence in injected rHGGs/rGBMs may have been due to absence of a robust anti-HSV1 immune response. Coupled with the extended survival for patients with positive HSV1 serology (Fig. 1c), this suggests that tumour clearance of CAN-3110 characterized patients with an improved survival response to CAN-3110.

## T cell metrics are linked to survival

The previous data (Fig. 2c) showed that CAN-3110 elicited an increased number of TILs in post-treatment samples that correlated with patient

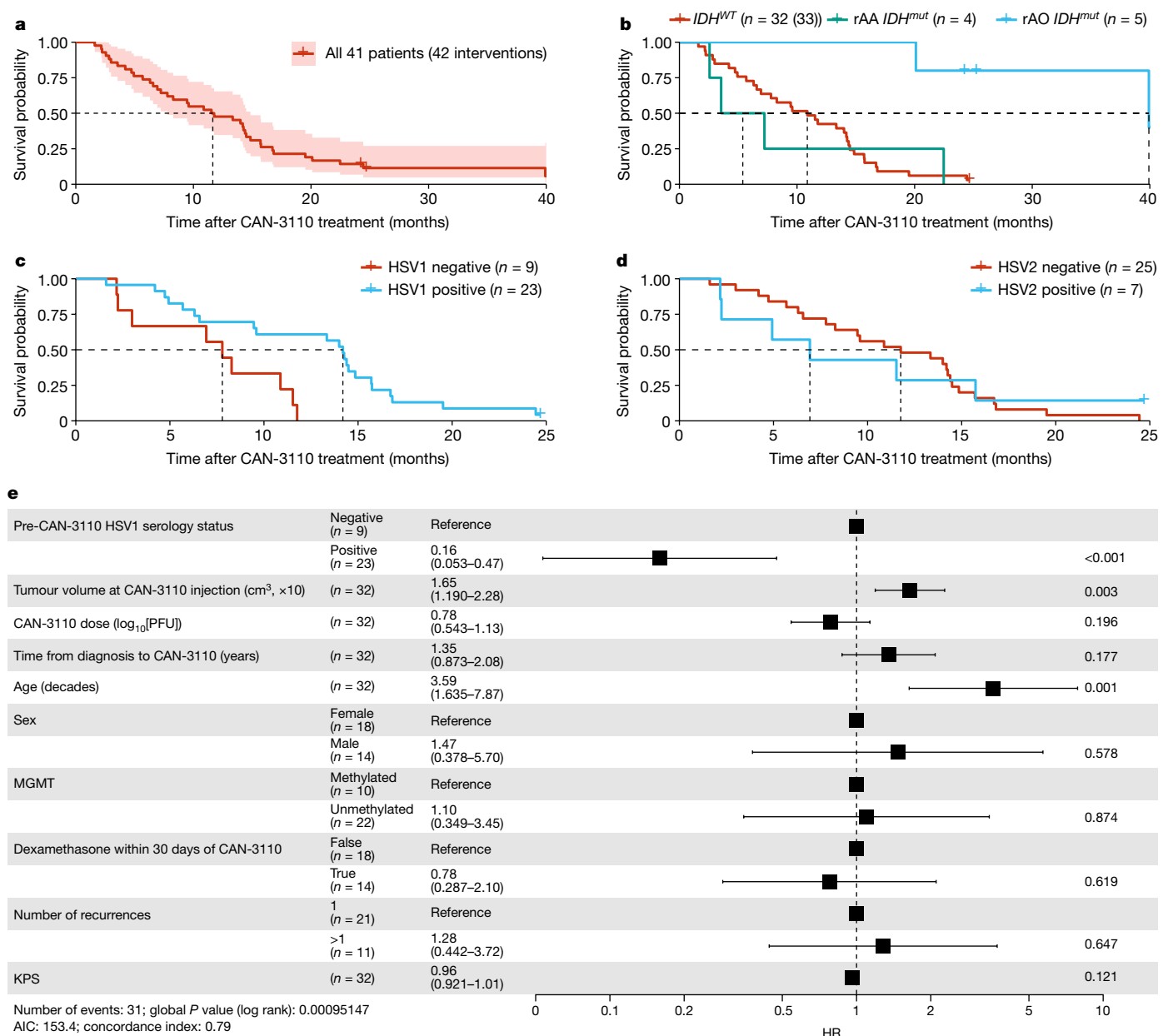

**Fig. 1 | Survival data. a**, Kaplan–Meier survival analysis of 41 patients with rHGG (42 interventions) after treatment with CAN-3110 (day 0). The shaded area shows the 95% CIs; the Kaplan–Meier estimate of survival probability is shown. Data maturity, October 2022. Median survival time (MST), 11.6 months (95% CI = 7.8–14.9 months). **b**, Kaplan–Meier survival analysis of patients with *IDH^{WT}* rGBM (*n* = 32 patients, 33 interventions), *IDH^{mut}*rAA (grades 3 and 4; *n* = 4 patients) and *IDH^{mut}* rAO (grade 3; *n* = 5 patients). MST, 10.9 months (*IDH^{WT}*rGBM; 95% CI = 6.9–14.4 months), 5.4 months (*IDH^{mut}*rAA; 95% CI = 2.6–∞ months) and 39.9 months (*IDH^{mut}* rAO; 95% CI = 39.9–∞ months). Hazard ratio (HR): *IDH^{mut}*rAO, 0.07 (95% CI = 0.01–0.49, *P* = 0.0079, two-sided Cox proportional-hazard test); *IDH^{mut}*rAA, 1.09 (95% CI = 0.38–3.16, *P* = 0.87, two-sided Cox proportional-hazard test). **c**, Kaplan–Meier survival analysis of 31 patients with *IDH^{WT}* rGBM (32 interventions) by negative (*n* = 9) or positive (*n* = 22 patients, 23 interventions) HSV1 serological status after treatment with CAN-3110. MST, HSV1 positive, 14.2 months (95% CI = 9.5–15.7 months); and HSV1 negative, 7.8 months (95% CI = 3.0–∞ months). *P* = 0.007 (two-sided likelihood ratio test). **d**, Kaplan–Meier

survival analysis of 31 patients with *IDH^{WT}* rGBM (32 interventions) by negative (*n* = 24 patients, 25 interventions) or positive (*n* = 7) HSV2 serological status before treatment with CAN-3110. MST, HSV2 positive, 6.9 months (95% CI = 2.2–∞ months); and HSV2 negative, 11.8 months (95% CI = 8.3–14.5 months). *P* = 0.9 (two-sided likelihood ratio test). **e**, Cox proportional-hazard ratio multivariate analyses for independent predictors of survival in patients with *IDH^{WT}* rGBM after treatment with CAN-3110. The error bars and values in parentheses show the 95% CIs. *P* values calculated using two-sided Cox proportional-hazard tests are shown on the right for each covariate. The unit of tumour volume is increments of 10 cm³. Partial MGMT promoter methylation was treated as unmethylated. For patients who were administered dexamethasone within the 30 days before or after CAN-3110 treatment, the median dose was 4 mg per day, and the median number of treated days during this time was 14.5 days. KPS, Karnofsky performance score. For **c**–**e**, participant 045 was excluded due to non-GBM mortality. PFU, plaque-forming units.

survival in the HSV1-seropositive patients. To further validate this finding, we examined whether survival was also correlated with changes in T cell clonotype metrics in tumour and/or peripheral blood mononuclear cells (PBMCs). Again, we focused the analyses on the *IDH^{WT}* rGBM

population: out of the 29 paired rHGGs/rGBMs, 21 were *IDH^{WT}* rGBMs (corresponding to 20 patients). T cell receptor β chain (TCRβ) DNA sequencing (DNA-seq) was performed on tumours and corresponding PBMCs collected at various timepoints after injection (range, 7–349

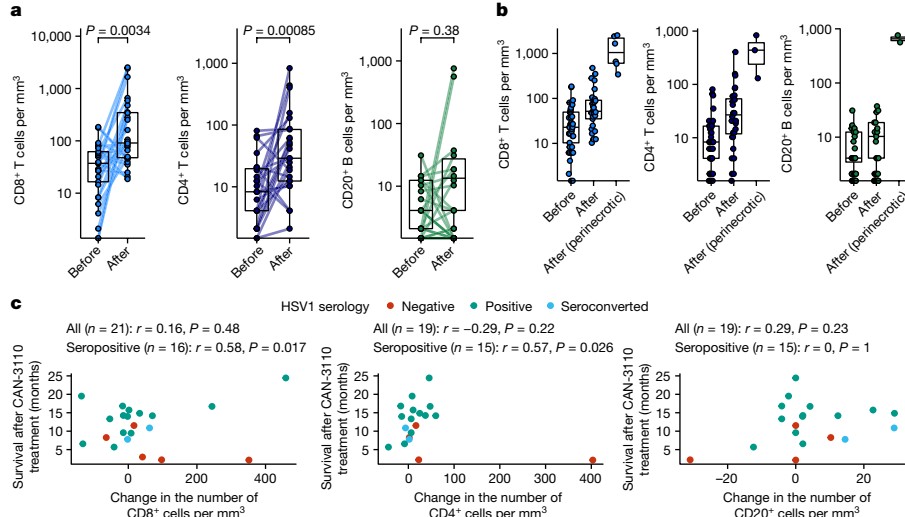

**Fig. 2 | Neuropathologic analyses. a**, Quantification of CD4+ and CD8+ T cells and CD20+ B cells from patients with available paired pre-treatment biopsies and post-treatment tumour samples distal from and/or directly adjacent to the virus injection site. *n* = 26 patients and 27 interventions (CD8+), and 24 patients and 25 interventions (CD4+ and CD20+). *P* values were calculated using two-sided Wilcoxon matched-pairs signed-rank tests. **b**, Quantification of CD8+, CD4+ and CD20+ cells in pre-treatment and post-treatment samples in tumour areas far from the CAN-3110 injection site versus tumour areas near to necrotic foci associated with CAN-3110 injection. For pre-treatment, post-treatment and perinecrotic areas, respectively, *n* patients (interventions) = 39 (40), 29 (30) and 6 (6) (CD8+); 37 (38), 29 (30) and 3 (3) (CD4+); 36 (37), 29 (30) and 2 (2) (CD20+). **c**, Correlations between changes in immune counts and post-treatment survival

for CD8+ (left), CD4+ (middle) and CD20+ (right) cells in *IDH^WT^* rGBMs. Pearson's correlation coefficient *r* and *P* values (two-sided, based on *t*-distribution) are provided above each plot calculated either using all patients or using only patients who were HSV1 seropositive before or after treatment. When counts were available for multiple post-treatment timepoints for a patient, the timepoint with the highest number of CD4+CD8+ cells was chosen. Importantly, TIL counts were not significantly confounded by the collection timepoint (Extended Data Fig. 7a–c). Patient 045 was excluded from the analyses in **c** due to early non-GBM mortality. The box plots show the median (centre line), 25th and 75th percentiles (box limits) and up to 1.5× the interquartile range or to the minimum/maximum values (if <1.5 × interquartile range distance from the box) (whiskers).

days). These data were used to calculate changes in the T cell fraction and metrics of TCRβ diversity (productive entropy and productive Simpson clonality; Supplementary Methods). Again, these metrics were not significantly confounded by the collection timepoint (Extended Data Fig. 9a). We found that changes in the tumour T cell fraction (a measure of T cell frequency) after CAN-3110 treatment were positively correlated with prolonged post-treatment survival both in tumours of all of the patients and in tumours of the patients who were HSV1 seropositive (Fig. 4a and Extended Data Fig. 9b). Increased tumour TCRβ diversity (increased entropy/decreased clonality) was associated with prolonged post-treatment survival both in tumours of all of the patients and in tumours of patients who were HSV1 seropositive (Fig. 4b and Extended Data Fig. 9c,d). The same findings were observed for PBMCs (Fig. 4c,d and Extended Data Fig. 9e), suggesting that evolution of a polyclonal T cell response was correlated with survival. Notably, the association between HSV1 serology status and survival was maintained in the subset of patients with *IDH^WT^* rGBM for which TCRβ sequencing data were available (Extended Data Fig. 9f). Tumours from patients positive for HSV1 had nominally higher productive entropy (that is, higher TCRβ rearrangement diversity) compared with those from patients negative for HSV1 after (*P* = 0.070) but not before (*P* = 0.65) CAN-3110 treatment (Extended Data Fig. 9g), suggesting that TCRβ diversity after CAN-3110 treatment was influenced by positive HSV1 serological status.

We also performed bulk RNA-seq analysis of a subset of *IDH^WT^* rGBMs for which tumours were frozen (to obtain good-quality RNA) and identified transcripts that possessed a V(D)J junction (indicating a T or B cell receptor transcript). The total number of pre-treatment V(D)J transcripts was significantly correlated with post-treatment survival, with a trend towards significance with total post-treatment V(D)J transcript counts (Extended Data Fig. 9h,i), whereas the numbers of unique V(D)J transcripts both before and after treatment were significantly correlated with survival (Extended Data Fig. 9j,k), further

validating the association between TCR abundance/diversity and post-treatment survival.

## Specific public T cells are linked to survival

We next examined whether there were specific T cell clonotypes that were associated with participant response to therapy. To do this, we focused on public T cell clonotypes[36], shared among the 21 *IDH^WT^* rGBMs for which we had TCRβ sequencing data. As expected, public TCRβs between patients were relatively rare in PBMCs and even more so in tumours (Extended Data Fig. 10a–f and Supplementary Methods). We found 55 public TCRβ sequences in 21 paired PBMC samples that we could analyse. There were highly significant changes in the frequency of two public PBMC T cell clones that were significantly associated with survival after treatment with CAN-3110: CASSLGGNTEAFF[37,38] (Extended Data Fig. 10g; false-discovery rate (FDR) = 0.0035) and CASSSSTDTQYF[39] ((Extended Data Fig. 10h; FDR = 0.018). Taken in conjunction, these findings show that survivorship after CAN-3110 treatment in the studied patients was significantly correlated with overall changes in T cell clonotype metrics and changes in the frequency of at least two specific public T cell clonotypes in PBMCs.

## Changes in T cell repertoire

Given the little overlap (very few public TCRs) in TIL-specific TCR clonotypes between patients (Extended Data Fig. 10), the relationship between survival after CAN-3110 treatment and TCR clonotype frequency changes could not be meaningfully analysed in TILs. There has been recent interest in analyses of tumour/PBMC T cell clonal repertoire changes as a function of oncologic immunotherapy[40]. Similarly, we sought to determine whether the tumour/PBMC T cell clonal repertoire changed after treatment with CAN-3110. We found 63 TCRs that were significantly (FDR ≤ 0.05) expanded or depleted in TILs of 11 of

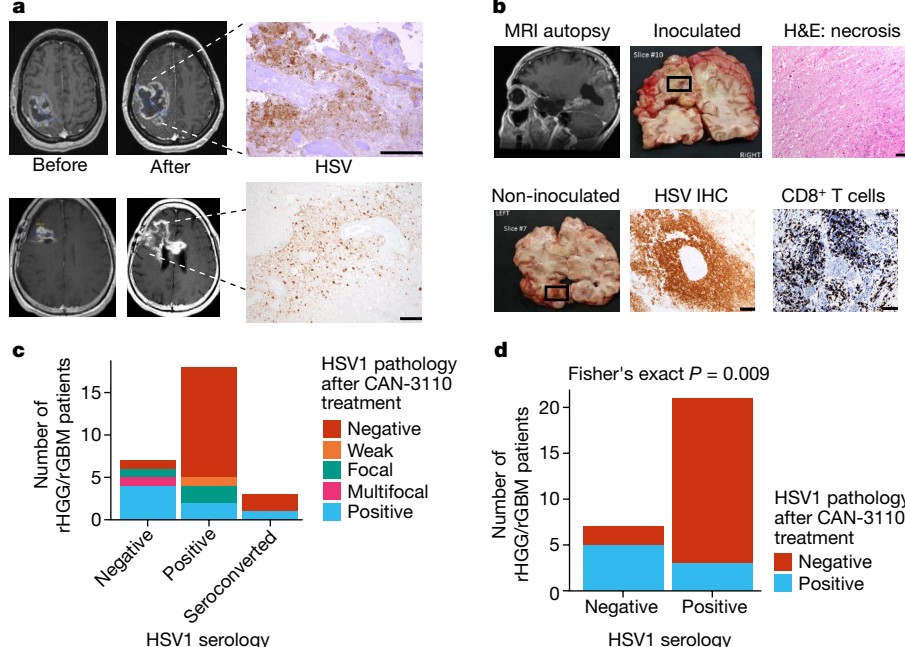

**Fig. 3 | CAN-3110 persistence in injected rHGG/rGBM is associated with negative HSV1 serological status either before or after therapy.**
**a**, oHSV-positive immunohistochemistry (IHC) images from two participants. Top, magnetic resonance imaging (MRI) images before and 41 days after CAN-3110 injection ($10^6$ PFU) from patient 005. oHSV-positive immunohistochemistry was visualized in the large area of tumour necrosis. The area was also positive for oHSV DNA as determined using PCR and positive for *ICP22* oHSV transcripts as determined using quantitative PCR with reverse transcription (RT–qPCR; data not shown). Bottom, MRI images from patient 028 before and 253 days after CAN-3110 injection ($10^9$ PFU). oHSV-positive immunohistochemistry images were visualized in the area of resected tumour necrosis; the positive status of *ICP22* oHSV transcript was determined using RT–qPCR (data not shown). **b**, Participant 014 had multifocal GBMs in the left temporal and left occipital lobes. The left occipital lobe lesion was injected with $10^7$ PFU of CAN-3110. Post-mortem analyses were performed 252 days after injection. Top left,

MRI scan before post-mortem brain collection, with the necrotic injected occipital lesion, shown in the grossly necrotic lesion (top middle), confirmed by histological haematoxylin and eosin (H&E) staining (top right). The CAN-3110 non-injected temporal-lobe post-mortem gross section (bottom left) exhibited oHSV positivity (bottom middle) and dense infiltrates of CD8+ T cells (bottom right). Extended Data Fig. 8 shows that this oHSV-positive focus was CAN-3110 and not reactivated latent wild-type HSV1 from this patient who was otherwise seronegative for HSV throughout the trial. **c**, HSV1 pathology staining in tumour tissue from patients with rGBM/rHGG ($n = 28$ interventions, 27 patients) after CAN-3110 treatment relative to HSV1 serological status. **d**, The same data as in **c**, but with patients who were initially seropositive grouped with patients who seroconverted after treatment with CAN-3110. Focal/weak pathology staining was grouped with negative staining; and multifocal staining was grouped with positive staining. *P* values were calculated using two-sided Fisher's exact tests. For **a** and **b**, scale bars, 100 μm.

the analysed patients with *IDH^WT* rGBM (Supplementary Table 3). If we looked at TCRs that concordantly changed in TILs and PBMCs, four TCRs significantly (FDR ≤ 0.05) expanded and five TCRs were significantly depleted in both TILs and PBMCs (Extended Data Fig. 11a). Of the four expanded TCRs common between TILs and PBMCs, three were from a single patient—patient 021—who was an exceptional responder after CAN-3110 treatment and remained radiologically tumour free for more than 2 years after CAN-3110 treatment before dying due to a non-GBM-related event (Extended Data Fig. 4b and Supplementary Video 1). Notably, all TCRs that concordantly expanded/depleted in both TILs and PBMCs were in longer-surviving patients (Extended Data Fig. 11b), suggesting that defined and concordant PMBC/TIL T cell clonal repertoire changes denoted responses after CAN-3110 treatment. In one participant (previously discussed in Fig. 3b and Extended Data Fig. 8) who remained HSV1 seronegative throughout the trial and was therefore unlikely to have T cell reactivity against HSV1, there were four expanding emergent T cell clonotypes (Extended Data Fig. 11c). This suggested that these were unlikely to be reactive against CAN-3110. When assessing V(D)J gene usage, we also identified a correlation between post-treatment TCRBV09-01*01 (refs. 41,42) usage and survival in HSV1-seropositive patients (Extended Data Fig. 11d; Pearson's *r* = 0.00019, FDR = 0.0095). Taken in conjunction, the analyses of T cell clonotypes in tumours revealed that longer-term survivors showed concordance between TIL and PBMC expansion, suggesting that there were alterations in the T cell repertoire after CAN-3110 treatment in

the patients who survived for longer. In at least one participant, there was suggestive evidence that tumour TCR expansion was unlikely to be against CAN-3110.

## Tumour immune signatures are linked to survival

We next queried RNA transcriptomic signatures in paired pre- and post-treatment frozen tumours (to maximize isolation of high-quality RNA) from 14 *IDH^WT* rGBMs (13 patients, 14 interventions). Notably, associations between post-treatment immune signatures and survival were stronger when analysing samples from only HSV1-seropositive patients compared with when analysing samples from all patients (Fig. 5a–c and Extended Data Figs. 12 and 13). In fact, analysis in HSV1-seropositive patients showed 13 post-treatment immune signatures associated with survival (Fig. 5b,c and Extended Data Fig. 13b), whereas, when analyses were conducted with all patients (HSV1 seronegative and seropositive), there were only 7 post-treatment signatures associated with survival (Fig. 5b and Extended Data Fig. 12b). Notably, most of the immune signatures in HSV1-seropositive patients became associated with survival only after treatment with CAN-3110 (Fig. 5c). The time to tumour collection after treatment did not influence the post-treatment signature analyses (Extended Data Fig. 12c). When considered together with other data from this study (Fig. 5d), these results demonstrate that CAN-3110 instigates a highly inflammatory and immunologically activated tumour microenvironment in HSV1 serologically positive

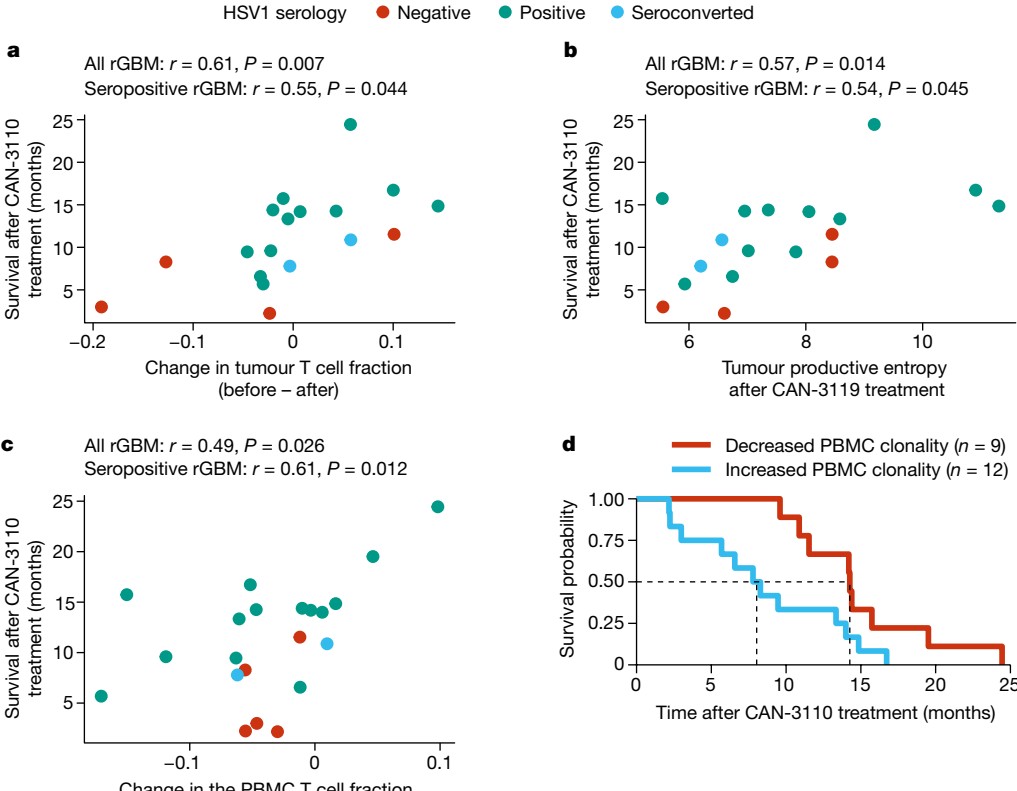

**Fig. 4 | TCR clonotype analyses. a**, The correlation between the change in tumour T cell fraction (after versus before CAN-3110 treatment) and survival after CAN-3110 treatment. The T cell fraction is the fraction of nucleated cells that are T cells on the basis of TCRβ DNA-seq analysis (see the 'Definition of TCR based metrics' section in the Supplementary Methods). **b**, The correlation between post-CAN-3110 tumour TCR productive entropy (Supplementary Methods) and survival. A higher entropy indicates a greater diversity of TCRβ rearrangements. $n = 18$ interventions and 17 patients. For **a** and **b**, three participants were excluded (two who survived longer than 1 year, and one who survived less than 1 year) with <200 ng of gDNA. $n = 18$ interventions and 17 patients. Extended Data Fig. 9b,c shows analyses with all patients, regardless

of the amount of gDNA collected. **c**, The correlation between the change in PBMC TCR clonotype fraction (after versus before CAN-3110 treatment) and survival after CAN-3110 treatment. $n = 21$ interventions and 20 patients. For **a**–**c**, Pearson's $r$ correlation coefficients and $P$ values (two-sided, based on $t$-distribution) are shown above the plots. **d**, Kaplan–Meier survival analysis based on an increase (change > 0) or decrease (change < 0) in PBMC productive Simpson's clonality (Supplementary Methods) after CAN-3110 treatment. $HR_{increased} = 2.79$ (95% CI = 1.08–7.21), $P = 0.034$ (two-sided Cox proportional-hazard test). Higher clonality indicates a lower diversity of TCRβ rearrangements.

patients that persists beyond detectable HSV1 antigen and is significantly correlated with post-treatment survival in a way that is not true of the pretreatment tumour immune state.

## Discussion

In this first-in-human clinical trial of CAN-3110, HSV meningitis or encephalitis was not seen, despite ongoing CAN-3110 persistence/replication for several months and maintenance of the *ICP34.5* neurovirulence gene. All inflammatory responses remained confined to injected tumours and were not detected in the surrounding brain tissue. This was true in HSV-seropositive and HSV-seronegative patients. Overall, CAN-3110 was well tolerated without dose-limiting toxicities.

A major challenge faced by solid tumour immunotherapy is to create a microenvironment that is favourable for an efficient immune response against cancer cells[43]. CD8[+] cytotoxic and CD4[+] helper T cells are important by expressing effector programs against tumour antigens. More recently, public (for example, the same TCR sequence is shared between different individuals) T cell clones, some of which recognize shared viral antigens, have also been shown to traffic into tumours, and their function in cancer immunity is a subject of debate[36]. In this trial, we analysed a large majority of paired pre- and post-CAN-3110 rHGG/rGBM tumours, with corresponding longitudinal PBMCs to show that

(1) pre-existing HSV1-positive serology correlated with individuals who survived the longest after treatment with CAN-3110; (2) CAN-3110 persisted in injected tumours, with almost half of assayed rHGGs still positive even months after a single timepoint injection, but persistence was significantly associated with negative HSV1 serology; and (3) CAN-3110 led to quantitative increases in TILs in a large majority of assayed tumours. Furthermore, we showed for the subpopulation with *IDH^WT* rGBM, for whom there were available paired tumour samples, (4) improved patient survival was correlated with changes in T cell clonotype metrics (elevated T cell clone frequency, increased TCRβ rearrangement diversity, decreased clonality in post-injection versus pre-injection tumours, and transcripts associated with immunological effector programs, particularly in the individuals seropositive for HSV1); and (5) there were changes in specific public peripheral TCR clonotypes significantly associated with survival after CAN-3110 treatment. Taken together, positive HSV1 serology with the observed changes in T cell clonotypes, including public ones, results in a more efficacious immune response, characterizing individuals whose immune system is more 'fit' and who can mount a more effective antiviral and possibly antitumour immune response. Note that two of the longest survivors were treated with immune-checkpoint inhibition after their injected tumours were resected (see the swimmer plots of participant 019 and 021 in Extended Data Fig. 3c,d), based on the post-injection

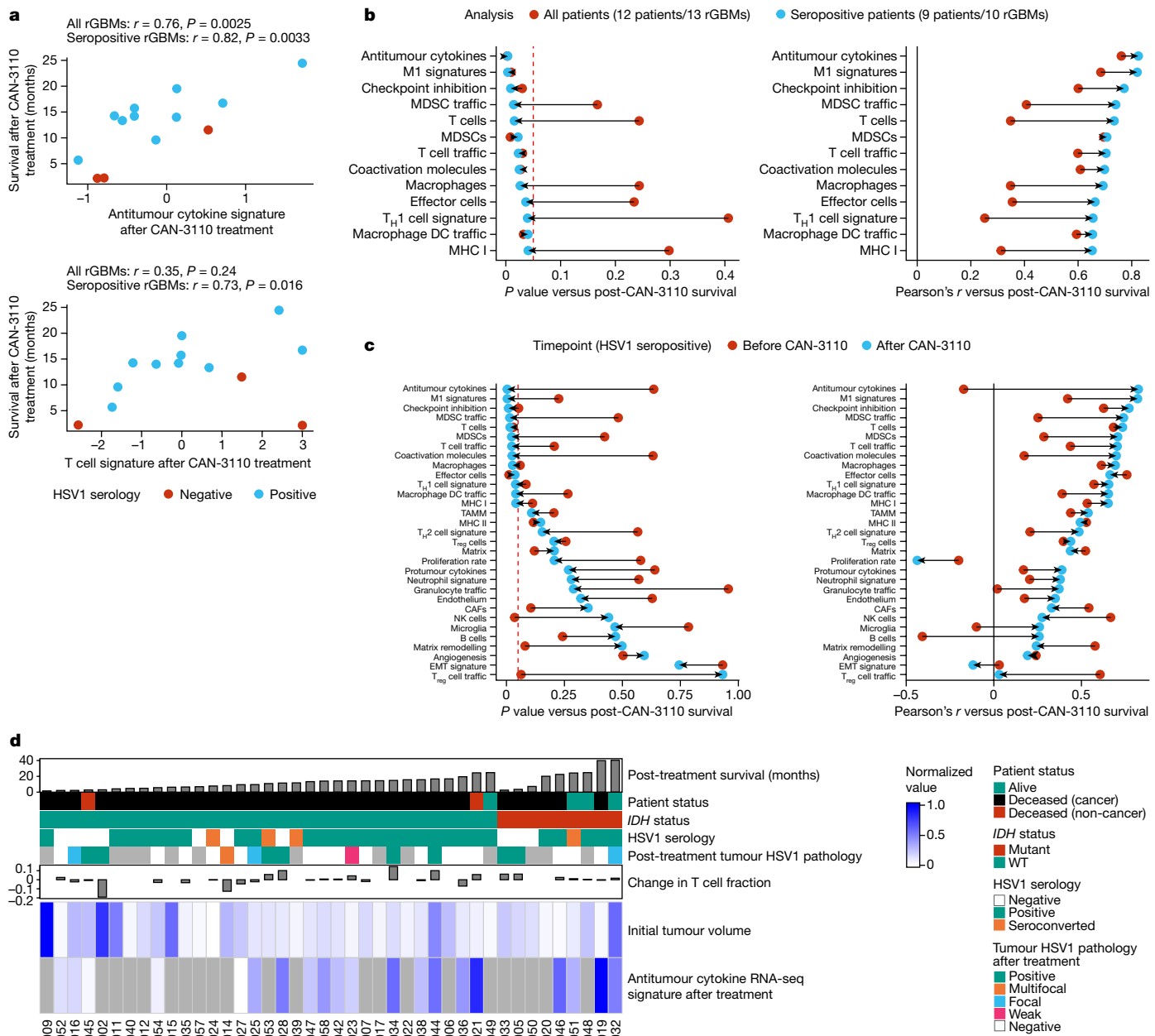

**Fig. 5 | Survival correlation between immune transcript signature programs in HSV1-seronegative and HSV1-seropositive patients.** A total of 13 paired *IDH*^WT rGBMs with good-quality RNA was analysed by bulk RNA transcriptomics. Transcriptomic signatures for different biological programs were estimated for each sample, and these signatures were assessed for correlation with survival after CAN-3110 treatment either in all patients or only in patients who were HSV1 seropositive before or after CAN-3110 treatment. **a**, Example of two immune signatures (antitumour cytokine and T cell signatures) that are strongly correlated with survival after CAN-3110 treatment when analysed in HSV1 seropositive patients. Pearson's *r* correlation coefficients and *P* values (two-sided, based on *t*-distribution) are shown above the plots. Importantly, these signatures did not appear to be significantly confounded by the tissue collection timepoint (Extended Data Fig. 12c). **b**, The change in Pearson's correlation *P* (left) (two-sided, based on *t*-distribution) and *r* (right) values when correlations between post-treatment immune signatures and survival were performed in all patients (red points) or in only HSV1-seropositive patients

(teal points). Only gene signatures that reached $P \le 0.05$ (dashed red line) in either analysis were plotted. DCs, dendritic cells; MDSCs, myeloid-derived suppressor cells; $T_H1$, T helper 1. **c**, The change in Pearson's correlation *P* (left) (two-sided, based on *t*-distribution) and *r* (right) values for pre-treatment (red points) and post-treatment (teal points) samples from HSV1-seropositive patients. This panel includes all of the analysed RNA-seq gene signatures. The dashed red line indicates $P = 0.05$. CAFs, cancer-associated fibroblasts; EMT, epithelial–mesenchymal transition; NK cells, natural killer cells; TAMM, tumour-associated monocyte/macrophage; $T_{reg}$ cells, regulatory T cells. **d**, Combined data for all of the patients in the study, including survival after CAN-3110 treatment, HSV1 serology, HSV1 tumour pathology, T cell fraction changes based on TCRβ DNA-seq, initial tumour volumes and bulk RNA-seq-based antitumour cytokine signature scores. The grey boxes indicate missing data. For **b** and **c**, HSV1 serology remained unchanged after CAN-3110 treatment for all of the patients, and one patient (045) was omitted from the analysis due to early non-GBM mortality.

finding of extensive TILs. We speculate that CAN-3110 inflamed the TME, possibly improving the efficacy of immune-checkpoint inhibition therapy.

The finding that positive HSV1 serology before or after CAN-3110 treatment was a highly significant independent predictor of response was unexpected based on previously reported trials of other

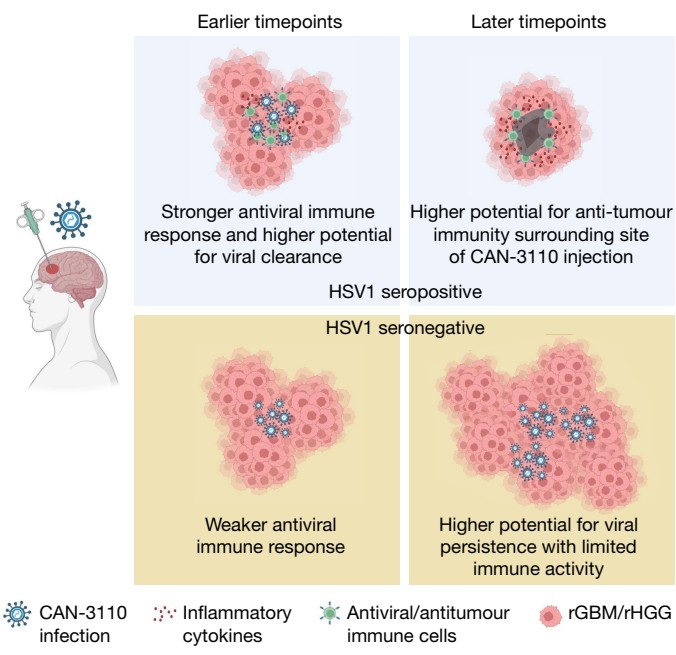

**Earlier timepoints**

**Later timepoints**

Stronger antiviral immune response and higher potential for viral clearance

Higher potential for anti-tumour immunity surrounding site of CAN-3110 injection

**HSV1 seropositive**

**HSV1 seronegative**

Weaker antiviral immune response

Higher potential for viral persistence with limited immune activity

❖ CAN-3110 infection ⁖ Inflammatory cytokines ✳ Antiviral/antitumour immune cells ● rGBM/rHGG

**Fig. 6 | A model for CAN-3110 action as a function of HSV1 serology.** In patients who are seropositive for HSV1, CAN-3110 elicits an initial augmented anti-HSV1 innate and T cell-mediated response (presumably by expansion and differentiation of memory into effector anti-HSV1 T cells) to clear the injected oHSV from tumours. This bystander T cell effect possibly mediates an effective antitumour effect by direct inflammation in the tumour and/or by stimulating 'antigen spreading' to also elicit T cell recognition of tumour antigens. In patients who are seronegative for HSV1, the absence of a rapid anti-HSV1 innate and T cell response leads to CAN-3110 replicative persistence with tumour growth overcoming viral-induced cytotoxicity and delayed immune activity against tumour antigens. The figure was generated using BioRender.com.

oHSVs[16,17,19,31]. A recent study showed no correlation between HSV1 serology in humans with GBM and survival[44]. We speculate that this finding may be specific to oncolytic viruses, based on the capacity of each oncolytic virus to replicate, persist and stimulate an innate and adaptive immune response. It may also be a factor related to sample size, at least for the brain tumour trials, as our trial had more participants. Note that the 22 participants (23 interventions) with *IDH*[WT] rGBM who were serologically positive for HSV1 before treatment with CAN-3110 had a mOS of 14.2 months (95% CI = 9.5–15.7 months; Fig. 1c), which is higher than the historical mOS of 6–9 months. Further prospective validation of this discovery in the next phase of planned trials will determine whether HSV1 serology can be used as a selection criterion for the likelihood of response.

The observation that CAN-3110 was immunohistochemically detected in almost half of the injected tumours several months (and even years in some patients) and even in one uninjected tumour suggests ongoing replication of the agent. Other oncolytic viruses, such as ICP34.5-defective oHSV, have rarely been found in injected human tumours, particularly after several weeks[17,19–21,31,45–47], suggesting that CAN-3110 expression of *ICP34.5* may enable persistence. We speculate that this persistence may increase infiltration of virus-specific TCR clones that could initially function in antitumour immunity in a bystander manner[36], but could also begin to stimulate T cell responses against tumour antigen. Mouse brain tumour models do show that tumour infiltration of T cells against both tumour and viral antigens correlate with survival[48]. The significant association of HSV1 seropositivity with the absence of CAN-3110 antigen and transcripts in tumours after injection suggests that an initial humoral and probably adaptive antiviral immune response led to an improved antitumour

response based on the survival data and on the finding that there were still increased CD8[+] and CD4[+] T cells and increased immunological transcriptional programs in tumours despite absent CAN-3110 in the longer-surviving patients (Fig. 6). Identification of the expansion of emergent TCRs, such as those in patient 014 who was seronegative for HSV1 before and after CAN-3110 treatment, possibly suggest that oHSV therapy indeed promotes epitope spreading[49], enabling expansion of T cell clones against tumour antigens. Future extensive studies determining whether the TCRs that we discovered in injected tumours react to viral versus tumour antigens are underway (data not shown).

In summary, single-timepoint intralesional injection of rHGG/rGBM with CAN-3110 enriches the tumour microenvironment with TILs, inducing defined changes in peripheral and tumour T cell repertoires and tumour transcriptomic signatures. These changes are particularly evident in patients who are seropositive for HSV1 and are associated with improved survival in this otherwise therapy-refractory cancer. These findings therefore provide human immunological and biological evidence supporting intralesional oncolytic modalities to convert the immunosuppressive TME characteristic of many solid cancers into a TME that is more favourable to immunologic rejection of the tumour. We are now set to determine whether multiple-timepoint injections lead to further improvements in this therapy (ClinicalTrials. gov: NCT03152318).

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

## Reporting summary

Further information on research design is available in the Nature Portfolio Reporting Summary linked to this article.

## Data availability

Patient responses, demographic information and safety outcomes, as well IHC quantifications and RNA-seq gene signature scores are available within the Article and its Supplementary Information. Raw RNA-seq and TCRβ DNA-seq files have been deposited in a controlled-access repository in the Database of Genotypes and Phenotypes (http://www.ncbi.nlm.nih.gov/projects/gap/cgi-bin/study.cgi?study_id=phs003378.v1.p1). Source data are provided with this paper.

## Code availability

Custom code used to perform analyses in this study have been deposited at OSF (https://doi.org/10.17605/OSF.IO/YBCG7).

**Acknowledgements** We thank all of the patients and families who participated in the trial, particularly participant 021 and her family. We acknowledge assistance from S. Ventz, G. Park, L. M. Seften, K. Hoe Chow, A. Shetty, W. Pisano, E. Lapinskas and F. Watkinson; D. Hamm and B. Banbury for assistance in TCR analyses; and N. Caffo, J. Dwyer, D. Lane and H. White for assistance.

**Author contributions** Conceptualization: E.A.C. Methodology: A.L.L., I.H.S., A.M.L., H.N., J.K.W., A. Santos, N. Masud, G.F., X.M., A.S.Y., J.G., A.Z., J.D.B., E. Torio, H.I., J.L., N. Shono, M.O.N., P.H., R.P., H.S., B.S., W.L.B., P.P., E.C., S.W.M., M.C.P., Y.Z., A.C., S.J.R., K.K., E. Tikhonova, N. Miheecheva, D. Tabakov, N. Shin, A.G., A. Shumskiy, F.F., E.A.-C., L.K.A., J.W., D. Krisky, A.M., C.M., P.P.T., F.B., D. Kovarsky, I.T., M.L.S., K.W.W., K.L., D.A.R. and E.A.C. Investigation: A.L.L., I.H.S., A.M.L., H.N., J.K.W., A. Santos, N. Masud, G.F., X.M., A.S.Y., J.G., A.Z., J.D.B., E. Torio, H.I., J.L., N. Shono, M.O.N., D. Triggs, P.H., R.P., H.S., B.S., W.L.B., P.P., S.W.M., M.C.P., Y.Z., A.C., S.J.R., P.Y.W., E.Q.L., L.N., U.C., L.N.G.C., S.D.D., T.B., K.K., E. Tikhonova, N. Miheecheva, D. Tabakov, N. Shin, A.G., A. Shumskiy, F.F., E.A.-C., L.K.A., D. Krisky, J.W., A.M., C.M., P.P.T., F.B., D. Kovarsky, I.T., M.L.S., K.W.W., K.L., D.A.R. and E.A.C. Visualization: A.L.L., I.H.S., A.M.L., H.N., A. Santos, G.F., X.M., N. Shono, S.W.M., M.C.P., E. Tikhonova, N. Miheecheva, D. Tabakov, N. Shin, A.G., A. Shumskiy, F.F., K.L. and E.A.C. Funding acquisition: A.L.L., J.K.W., N. Shono, E.A.-C., P.P.T., M.L.S., K.W.W., K.L., D.A.R. and E.A.C. Project administration: H.N., S.J.R., P.Y.W., T.B., N. Shin, E.A.-C., P.P.T., F.B., I.T., M.L.S., K.W.W., K.L., D.A.R. and E.A.C. Supervision: I.H.S., H.N., G.F., X.M., S.J.R., P.Y.W., T.B., N. Shin, P.P.T., F.B., E.A.-C., L.K.A., M.L.S., K.W.W., I.T., K.L., D.A.R. and E.A.C. Writing—original draft: A.L.L. and E.A.C. Writing—review and editing: A.L.L., I.H.S., A.M.L., H.N., J.K.W., G.F., X.M., J.D.B., M.C.P., P.Y.W., L.N.G.C., N. Shin, E.A.-C., L.K.A., P.P.T., F.B., D. Kovarsky, M.L.S., K.W.W., D.A.R. and E.A.C.

**Competing interests** J.D.B. has an equity position in Treovir, an oHSV clinical stage company and is a member of the POCKiT Diagnostics board of scientific advisors; and has a provisional patent (application number 63/273,577) entitled 'Methods and formulations related to the intrathecal delivery of oncolytic viruses'. W.L.B. consulted for Stryker. P.P. is cofounder and member of the board of directors of Ternalys Therapeutics; he is also named as an inventor on patents related to non-coding RNA technology. P.Y.W. received research support from AstraZeneca/Medimmune, Beigene, Celgene, Chimerix, Eli Lily, Genentech/Roche, Kazia, MediciNova, Merck, Novartis, Nuvation Bio, Puma, Servier, Vascular Biogenics and VBI Vaccines and served on advisory boards for AstraZeneca, Bayer, Black Diamond, Boehringer Ingelheim, Boston Pharmaceuticals, Celularity, Chimerix, Genenta, GlaxoSmithKline, Karyopharm, Merck, Mundipharma, Novartis, Novocure, Nuvation Bio, Prelude Therapeutics, Sapience, Servier, Sagimet, Vascular Biogenics and VBI Vaccines, and on data safety monitoring committees for Day One Bio and Novocure. L.N. serves as a consultant for Ono and Brave Bio. L.N.G.C. received research support from Merck (to the Dana-Farber Cancer Institute); he also has received research support from the NIH, the American Society of Clinical Oncology and the Robert Wood-Johnson Foundation. E.Q.L. receives royalties from Wolter Kluwer (Up to Date) and consulting fees from GCAR. T.B. receives clinical trial support from ONO Pharmaceuticals, publishing royalties from UpToDate and Oxford University Press. S.J.R. receives research support from Bristol Myers-Squibb and KITE/Gilead; and is on the scientific advisory board for Immunitas Therapeutics. K.K. is employed by and owns equity in Clearpoint. E. Tikhonova, N. Miheecheva, D. Tabakov, N. Shin, A.G., A. Shumskiy and F.F. are employed by BostonGene and have equity options in BostonGene. E.A.-C. is a founder, board member of and holds equity in Candel Therapeutics. L.K.A. is co-founder and holds equity in Candel Therapeutics. D. Krisky, J.W., A.M., C.M., P.P.T. and F.B. are employees of and hold equity in Candel Therapeutics. I.T. is an advisory board member of Immunitas Therapeutics. M.L.S. and K.W.W. are equity holders, scientific co-founders and advisory board members of Immunitas Therapeutics. D.A.R. is an advisor to Agios, AnHeart Therapeutics, Avita Biomedical, Blue Rock Therapeutics, Bristol Myers Squibb, Boston Biomedical, CureVac, Del Mar Pharma, DNAtrix, Hoffman-LaRoche, Imvax, Janssen, Kiyatec, Medicenna Therapeutics, Neuvogen, Novartis, Novocure, Pyramid, Sumitomo Dainippon Pharma, Vivacitas Oncology and Y-mabs Therapeutics. E.A.C. is an advisor to Amacathera, Bionaut Labs, Genenta, Insightec, DNAtrix, Seneca Therapeutics and Theravir; he has equity options in Bionaut Laboratories, DNAtrix, Immunomic Therapeutics, Seneca Therapeutics and Ternalys Therapeutics; he is co-founder and on the board of directors of Ternalys Therapeutics. Patents related to oHSV and CAN-3110 are under the possession of Brigham and Women's Hospital with E.A.C. and H.N. named as co-inventors. These patents have been licensed to Candel Therapeutics. Present and future milestone license fees and future royalty fees are distributed to Brigham and Women's Hospital from Candel. The other authors declare no competing interests.

**Additional information**
**Correspondence and requests for materials** should be addressed to E. Antonio Chiocca.

## a

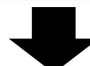

Subjects with frozen biopsy confirmation of recurrent glioma (previous diagnosis: high grade glioma (HGG)

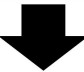

*Dose escalation (cohorts 1-9) (n=30)*
- Single timepoint stereotactic intraoperative MRI-guided injection of CAN-3110 into one site of recurrent HGG
- 3+3 dose escalation design
- doses of $1\times10^6$ to $1\times10^{10}$ PFU in half-log increments

*Dose expansion (cohort 10) (n=12)*
- Single timepoint stereotactic intraoperative MRI-guided injection of CAN-3110 into up to 5 sites of recurrent HGG.
- dose of $1\times10^9$ PFU

## b

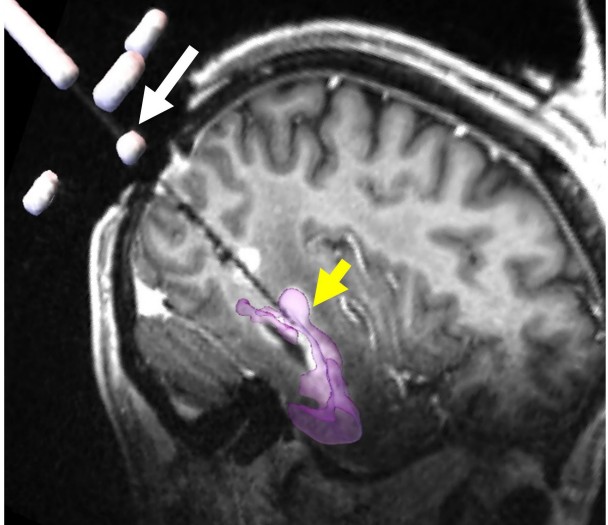

## c

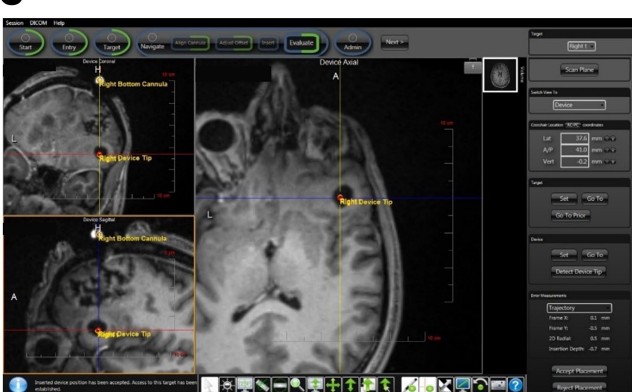

**Extended Data Fig. 1 | (related to Sub-heading, *Safety of CAN-3110 in patients with rHGG/rGBM*) Clinical trial design and treatment strategy. (a)** Dose-escalation schema- Subjects with a previous diagnosis of rHGG (Glioblastoma, Grade IV or III astrocytoma or anaplastic astrocytoma, grade III anaplastic oligodendroglioma, including molecular grading with or without a mutation in IDH and with or without hypermethylation of the MGMT promoter) were eligible for the trial. At the time of stereotactic biopsy, the neuropathologist had to confirm that there was histologic evidence consistent with glioma to exclude inclusion of subjects with radiation necrosis and/or infection. The first 9 cohorts of subjects underwent one stereotactic inoculation of CAN-3110 at a tumour site selected to be different from the antecedent biopsy site (to avoid blood contamination of the injectate) in a 3 + 3 dose-escalation design, starting from $10^6$ pfus up to $10^{10}$ pfus in half-log increments. The biopsy and injections for each subject were carried out in an intraoperative MRI to visualize injections in gadolinium-enhancing tumour. The volume of injectate was 1 ml delivered over 5 min using the SmartFlow cannula (ClearPoint Neuro, Inc.) that minimizes reflux. When all 30 subjects in the first 9 cohorts were treated (September 2017-February 2020) without a dose limiting toxicity, the protocol was amended to include a tenth cohort of 12 subjects, where up to 5 regions of tumour were injected with a dose of $10^9$ pfus divided into 1 to 5 mls based on tumour diameter (e.g., for each mm of tumour diameter, 1 ml of CAN-3110 was injected). No DLTs were encountered. Subjects in cohort 10 were accrued from June, 2020 until January, 2021. **(b-c)** Representative intraoperative MRIs during (**b**) and after (**c**) CAN-3110 injection. **b:** Subject 021 was positioned prone in the intraoperative MRI. The SmartFlow cannula is shown in the occipital area penetrating the skull through a drilled burr hole (white arrow). The T2 dark area shows the needle trajectory through the occipital and temporal lobe to reach the area of rGBM where the tip of the needle (yellow arrowhead) is placed for injection. The gadolinium-enhanced tumour was manually overlaid with purple colour. **c:** Representative intraoperative MRI (subject 002) showing the view from the intraoperative console after injection of CAN-3110 ($10^6$ pfus/1 ml). The T1 dark injectate is indicated by the blue and yellow cross-hatch, with the red dot showing where the tip of injection needle was after injection and needle removal, showing persistence of the injectate at site of injection with minimal reflux. The rGBM consisted of a bifrontal mass and the needle was inserted from the frontal vertex to reach an area of gadolinium-enhancement located in the inferior frontal lobe. The 3 images shown in the console are from the same brain section in coronal, sagittal and axial planes.

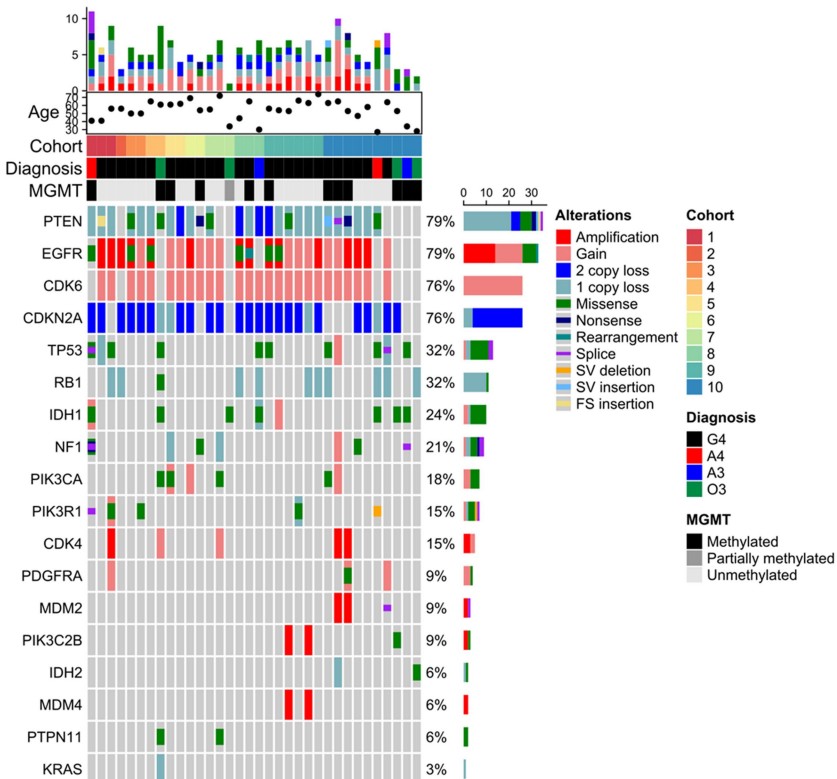

**Extended Data Fig. 2 | (related to Sub-heading,** *Safety of CAN-3110 in patients with rHGG/rGBM***) Genomic alterations in 34/41 subjects (42 separate interventions) in the CAN-3110 clinical trial.** 34 rHGG/rGBM specimens underwent exome sequencing for the 18 genes shown on the left of the panel (PTEN, EGFR, etc). On the right side of the panel, the percentage of tumours expressing each genetic mutation is listed together with colour coded relative frequencies of specific types of genomic alteration (amplification, gain, etc).

The top of the map displays a bar graph representation of tumour mutational burden (limited to these 18 genes), as well as indications of age, trial cohort, diagnosis at time of injection, and MGMT promoter methylation status for each patient included. To the far right of the figure, colour coding legends indicate designations for different types of genomic alteration, trial cohort, diagnosis, and MGMT promoter methylation status.

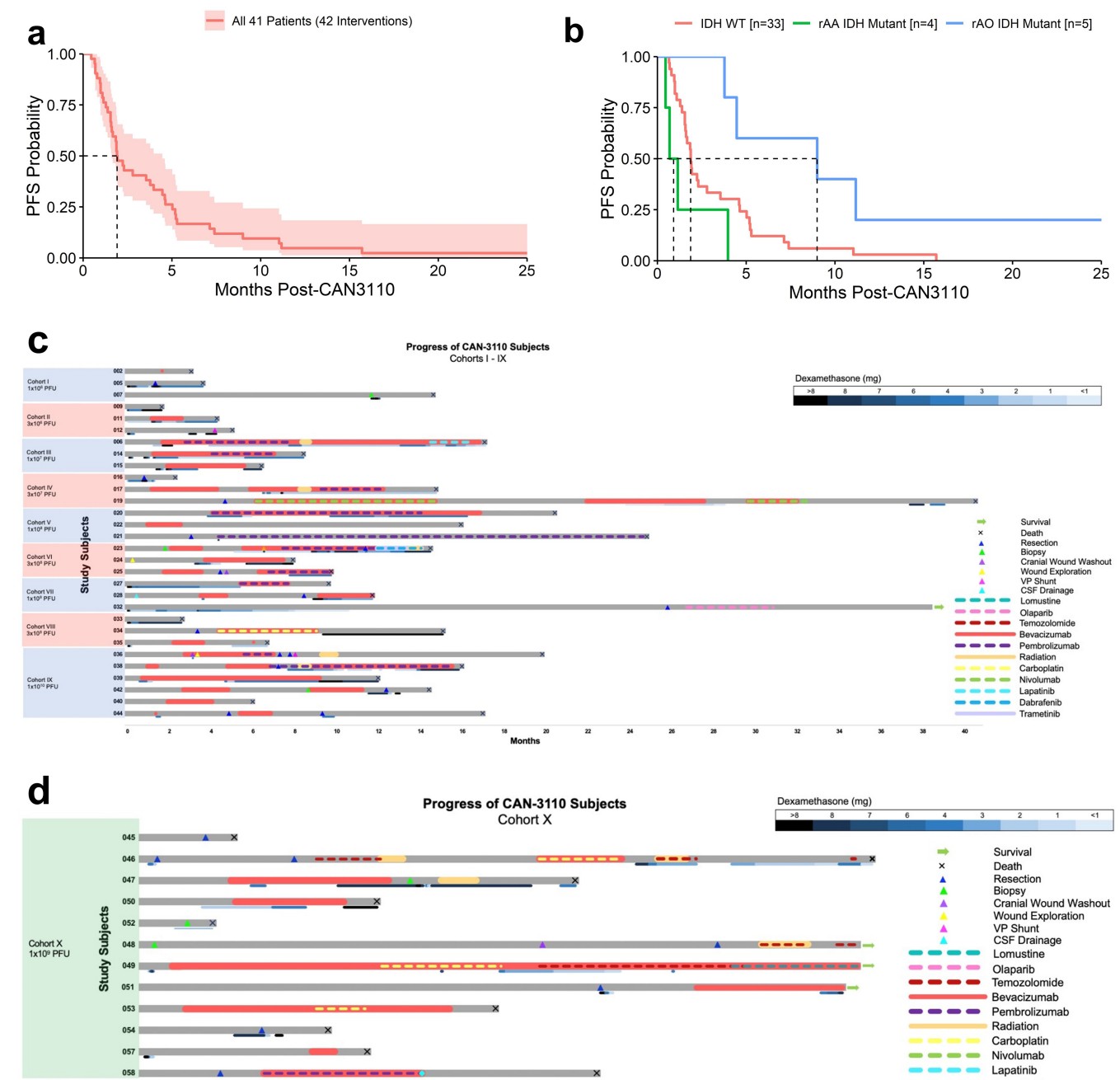

**Extended Data Fig. 3 | (related to Sub-Heading, *HSV1 serology predicts efficacy*). Trial Survival Outcomes.** (a,b) Progression-free survival for the entire rHGG/rGBM group (a) and the 3 rHGG sub-groups divided by rGBM IDHwt, rAA IDHmut, and rAO IDH mutant (b), based on the 2021 WHO classification of central nervous system tumours[5]. **(a)** Kaplan Meier progression free survival (PFS) for all 41 subjects (42 interventions). Shaded region = 95% confidence interval of KM estimate of survival probability. The data is mature as of October 2022. The median PFS was 1.9 months (95% CI: 1.6 – 4.5). **(b)** Kaplan Meier PFS curves for 41 subjects, divided into rGBM IDH wild type (n = 32 subjects, 33 interventions), rAA IDH mutant (n = 4), and rAO IDH mutant (n = 5). The median PFS were 1.9 (95% CI: 1.6 – 4.6), 0.9 (95% CI: 0.5 - Inf) and 9.0 months (95% CI: 4.5 - Inf), respectively. The PFS HR for rAO IDH mut was 0.30 (95% CI, 0.10 – 0.86, p = 0.026 (CoxPH, 2-sided)) and PFS HR for rAA IDHmut was 2.63 (95% CI: 0.91 – 7.65, CoxPH p = 0.075 (CoxPH, 2-sided)). **(c, d)** Swimmer plots for Cohorts 1-9 (i.e., I-IX) **(c)** and Cohort 10 **(i.e., X) (d)**. All 41 subjects' (42 interventions, with subject 042/054 treated twice, 6 months apart) clinical course since day 0

CAN-3110 injection time is shown. Cohort number and CAN-3110 dose are indicated on the far left, with next column showing each subject clinical trial ID. Months after CAN-3110 injection is shown below. After CAN-3110 injection, subjects were followed and when there was MRI evidence of progression or pseudoprogression with or without clinical deterioration, additional treatments were instituted including craniotomy and/or biopsy. All instituted treatments (bevacizumab, immune checkpoint inhibitors, carboplatin, temozolomide, reirradiation, lomustine, LITT, targeted inhibitors) have shown no benefit in this setting in advanced trials and were used for palliative purposes. Post-CAN-3110 treatments are shown in colour coding on the far right and time/duration of treatment is overlaid on the swimmer plot for each subject. Below each bar, colour coding for dexamethasone dosing and duration is shown for each subject. As of September 2022, there are 4 surviving subjects (032, 048, 049, 051), 3 of which are IDH mutant anaplastic oligodendroglioma (1p/19q co-deleted) and one is IDH wt GBM (049).

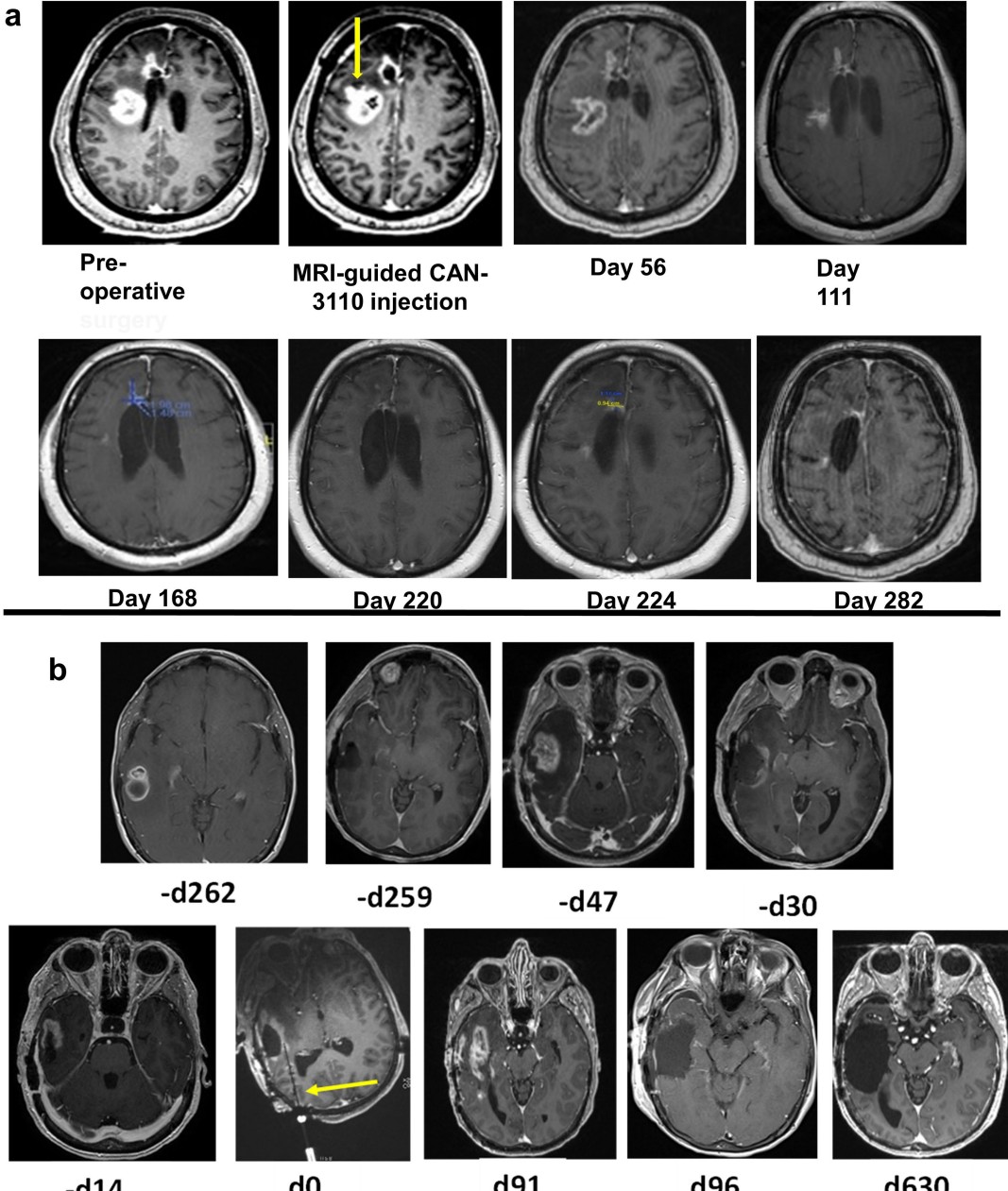

**Extended Data Fig. 4 | (related to Sub-Heading, *HSV1 serology predicts efficacy*) MRI imaging responses to CAN 3110. (a)** Complete response in a multifocal GBM subject. Subject 007 (56 year old caucasian man, IDHwt GBM) had an initial right frontal GBM resected. After completion of standard of care radiochemotherapy, the right frontal lesion grew back and a second new lesion posterior and periventricular also appeared (Pre-operative MRI). The subject underwent injection of CAN-3110 (10⁶ pfus in 1 ml) solely into the second new lesion (indicated by yellow arrow in MRI-guided CAN-3110 injection label). Serial MRIs on day 56, 111, 168, 220, 224 and 282 are shown. No other treatments and no dexamethasone were administered during this time, during which the patient experienced full time employment, travel and enjoyment from significant family events. At the 349 day mark, a new separate biopsy-proven recurrence in the right basal ganglia leading to a progressive hemiparesis and hemiplegia prompted the subject to seek hospice care and eventual demise. **(b)** Durable response in a right temporal GBM subject. Subject 021 (61 year old caucasian female, IDHwt GBM) had an initial GBM diagnosed 262 days (-d262)

before CAN-3110 injection. After craniotomy and tumour resection (-d259), she underwent standard chemoradiation and then treatment with temozolomide for IDHwt GBM with methylated MGMT promoter. The tumour recurred (-d47) and she underwent a second subtotal resection (-d30), but because of visible rapid progression (-d14), she was enrolled in the CAN-3110 trial. On d0, she received single injection of 10⁸ pfus (the MRI-compatible injection needle is indicated by the yellow arrow). On d91, MRI appeared to show progression and she was brought back to surgery for resection of the mass with postoperative MRI showing a gross total resection (d96). Histology and immunohistochemical staining showed a mixture of CD8+, CD4+, CD20+ lymphocytes and tumour (see Extended Data Fig. 6d, **panels labelled with #21)**. The subject then remained tumour free for the next 630 days (d630), which was the time of her last MRI. Unfortunately, she passed as the passenger of a motor vehicle accident on d717. The subject's personal story is shared in the supplementary video 1 with consent of her family.

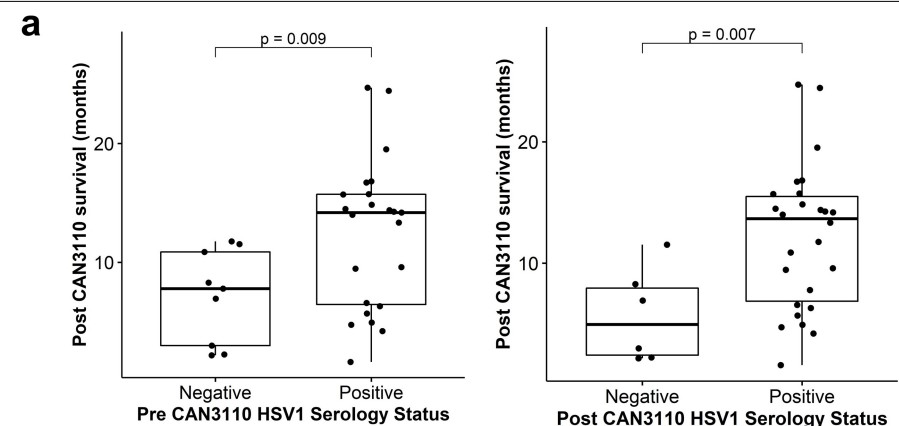

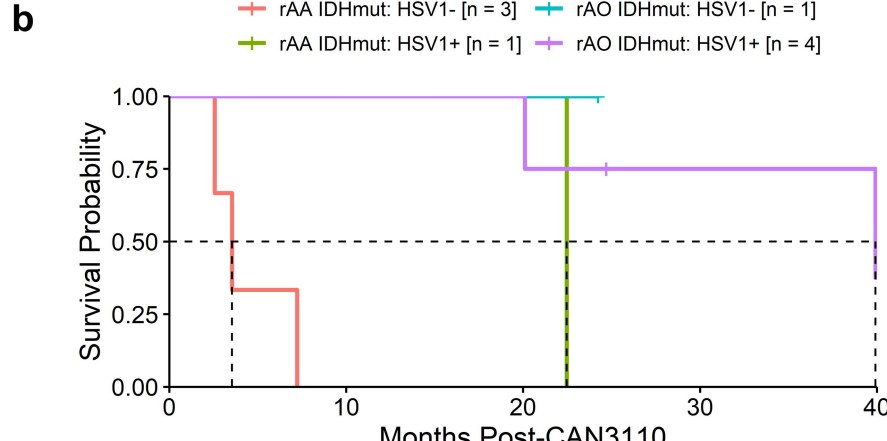

**Extended Data Fig. 5 | (related to Sub-Heading, *HSV1 serology predicts efficacy*). (a)** Comparative analyses of differential survival based on HSV1 pre-operative (left panel) and post-operative (right panel) serology for the 31 rGBM IDHwt subjects (32 interventions). P value from two-tailed Student's t-test. **(b)** Kaplan-Meier survival curves for IDH mutant patients as divided by pre-treatment HSV1 serological status. Median OS months with 95% confidence intervals = rAA HSV1−: 3.5 [2.6 - Inf]; rAA HSV1+: 22.5 [-Inf - Inf]; rAO HSV1−: not reached; rAO HSV1+: 39.9 [20.1 - Inf]. Boxplot centre = median, box bounds = 25th and 75th percentiles, whisker length = up to 1.5x inter-quartile range (IQR) or to minima/maxima (if <1.5x IQR distance from box).

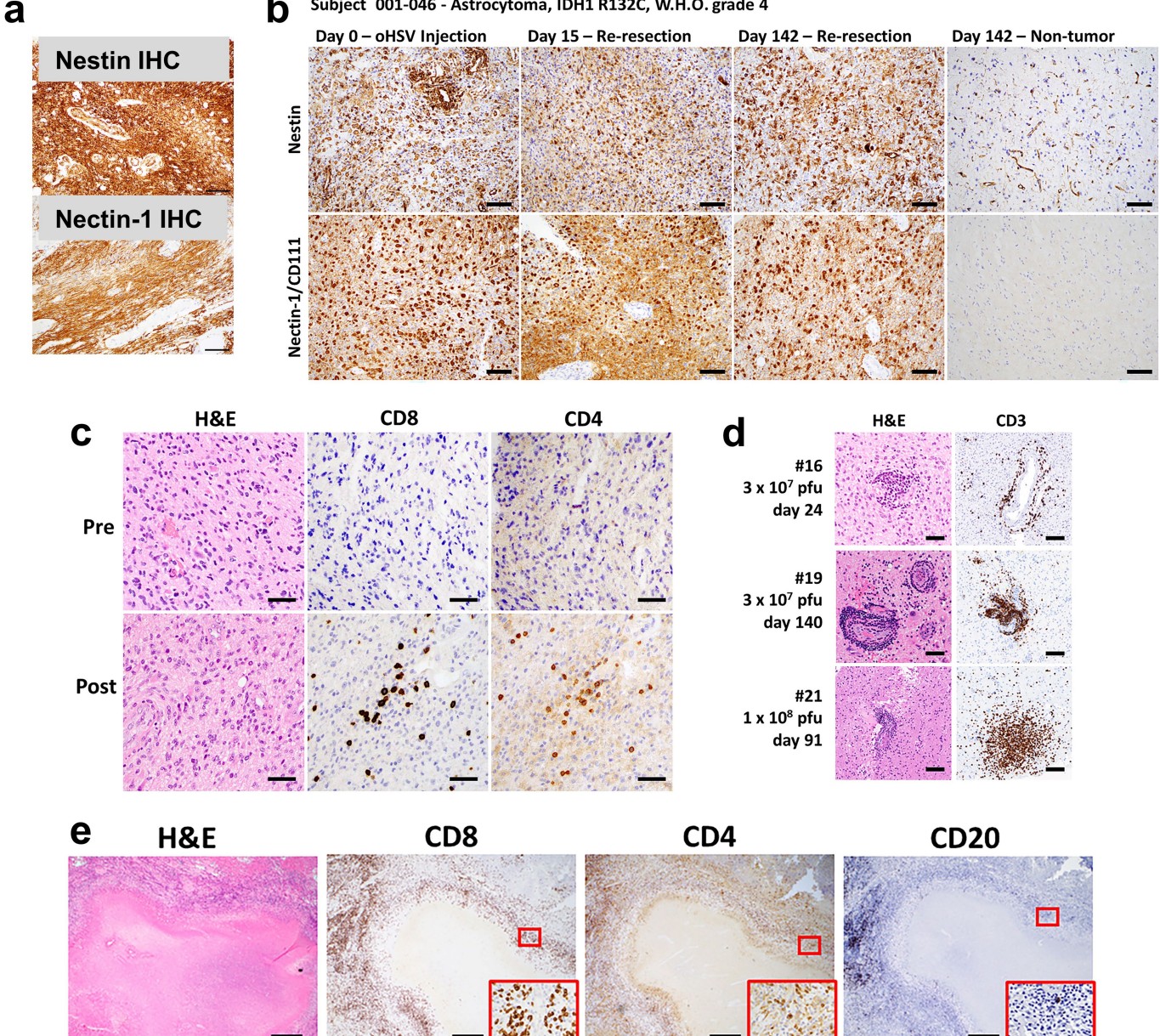

**Extended Data Fig. 6 | (related to Sub-heading, *CAN-3110 increases T cells in tumours*.) Representative immunohistochemistry (IHC) Images. (a)** upper panel: Nestin IHC was carried out in a rGBM (IDHwt) resected 279 days after CAN-3110 injection (Subject 044, 10^10 pfus in 1 ml); lower panel: Nectin-1/CD111 was carried out in a rGBM (IDHwt) resected 253 days after CAN-3110 (Subject 028, 10^9 pfu in 1 ml). **(b)** Time course of Nestin and Nectin-1 IHC. Subject 046 (IDHmut astrocytoma) was injected on day 0. Because of an SAE consisting of multiple seizures 2 days after injection (Table 1b), he was treated with antivirals with resolution of the event. He was then brought back for re-resection twice showing both times persistence of both nestin and nectin-1 expression with tumour progression. Non tumour brain shows nestin expression peri-vascularly with no nectin-1 expression. **(c)** Representative example of CD8+ T cell immunohistochemistry (IHC) from Pre- and Post-CAN-3110 injected tumour. Subject 016 was injected with $3 \times 10^7$ PFUs of CAN-3110 and post-injection tumour was resected 24 days after injection due to MRI evidence of continued progression. Areas shown were relatively far from the area of injection. **(d)** Representative examples of perivascular CD3+ T cell accumulation from 3 subjects (016, 019, 021) after CAN-3110 injection. Dose and time of post-injection tumour harvest are indicated for each. **(e)** Representative examples of perinecrotic accumulation of CD20+ B and CD8+ and CD4+ T cells from subject 034. Zoomed images of regions outlined in red are shown in bottom right corner of respective plots. Scale bars are 25 μm (panel **c**), 100 μm (panels **a**, **b**, and **d**), and 500 μm (panel **e**).

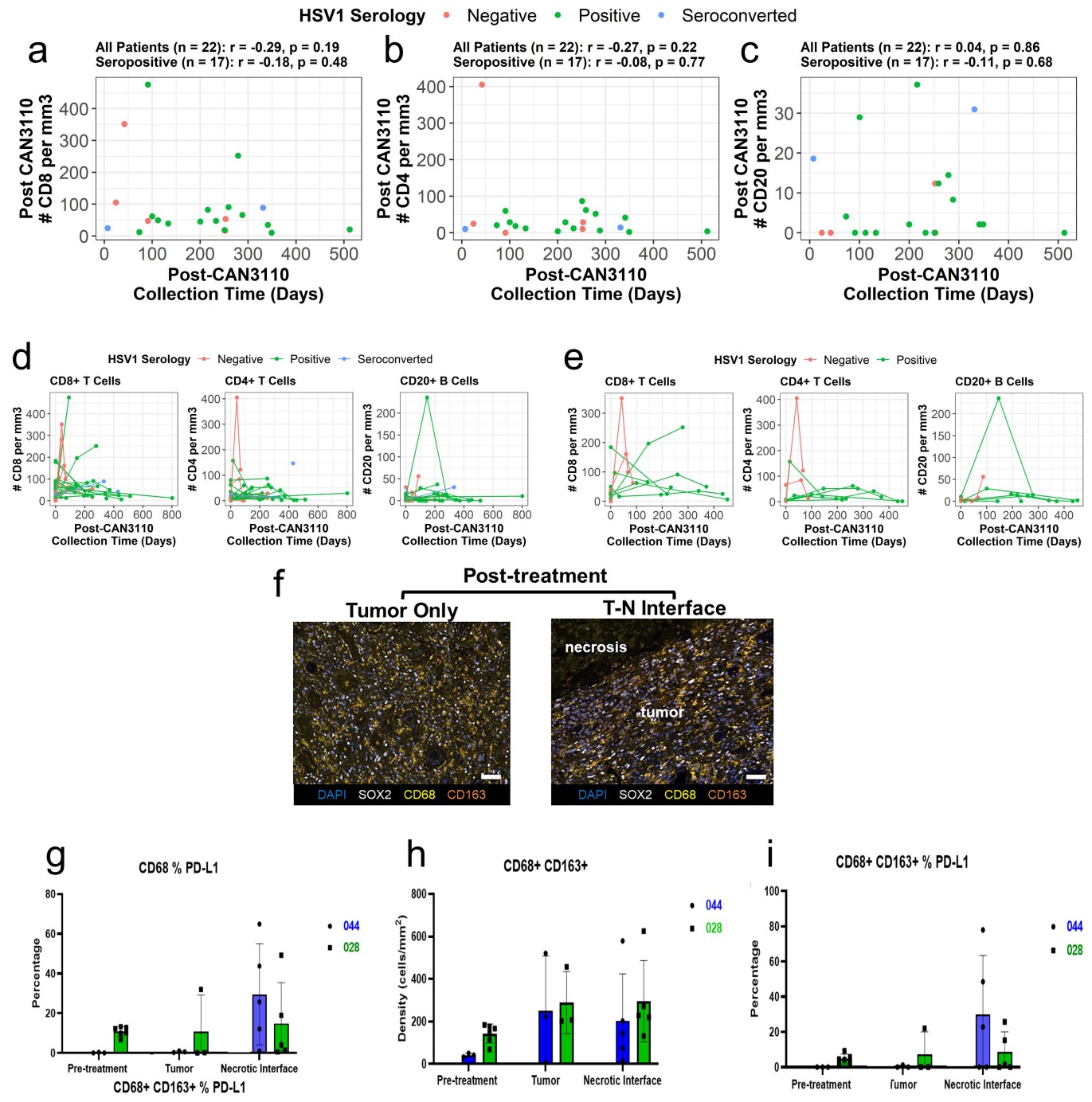

**Extended Data Fig. 7** | See next page for caption.

**Extended Data Fig. 7 | (related to Sub-heading, *CAN-3110 increases T cells in tumours*). Quantitative IHC.** IHC was performed by computer aided quantification of IHC stains (e.g., CD4, CD8, CD20) using slides scanned at 40X magnification using the Hamamatsu Nanozoomer S210. Using the Halo Image Analysis Sofware (PerkinElmer), 3 square regions of interest (approximately 160,000 mm$^2$ each) are averaged for each case in in areas of tumour and in uninvolved/reactive brain tissue if present, and quantities are normalized by tissue area (mm$^2$). **(a-c)** Pathological assessments of CD8+, CD4+, and CD20+ TILs are not systematically confounded by collection timepoint. Post-treatment IHC based counts of CD8+ (**a**), CD4+ (**b**), and CD20+ (**c**) TILs plotted versus the time of post-treatment tissue collection for the same IDHwt rGBM patients plotted in Fig. 2c (note that patient 045 was excluded due to early non-GBM mortality). Pearson's correlation coefficient r and p values (2-sided, based on t-distribution) are provided above each plot calculated either using all patients or using only patients which were HSV1 seropositive before or after treatment. When counts were available for multiple post-treatment timepoints for a patient, the timepoint with the highest number of CD4+ or CD8+ or CD20+ cells were chosen. **(d,e)** Quantitative IHC for CD8+, CD4+ T cells and CD20+ B cells for each patient as a function of time. For each subject, the number of CD8+, CD4+ T and CD20+ B cells/mm$^2$ are plotted as a function of time after CAN-3110 (i.e., when tumours underwent re-resection(s) and/or postmortem analyses after CAN-3110). Note that perinecrotic counts are not included here as they were only available for a few patients. n patients = 41. Kruskal-Wallis p = 0.33 (all patients), p = 0.16 (HSV1 seronegative patients), p = 0.45 (HSV1 seropositive/seroconverted patients) for CD8+. In **(e)**, the same data is shown as in panel **d** but restricted to patients that have >1 post-treatment sample available (i.e., underwent more than 1 resection or had a resection and then also a postmortem analysis). n patients = 8. Kruskal-Wallis p = 0.39 (all patients), p = 0.30 (HSV1 seronegative patients), p = 0.44 (HSV1 seropositive/seroconverted patients) for CD8+. **(f-i)** Multiplex fluorescent imaging (mIF) for myeloid cell populations in pre- and post-CAN-3110 rGBM IDHwt. Each 20x region of interest (ROI) is plotted as a black dot. The overlaying bar graph is the mean of the ROIs and the error bars represent the standard deviation. For panels **g** and **i**, the values represent the percentage of the macrophage populations (CD68+ for **panel g** and CD68+ CD163+ for **panel i**) that are positive for PD-L1 expression. For panel **h**, the values are the cell density, or number of positive cells per mm2 of CD68+ CD163+ double positive cells. **(f)** Representative mIF images of post-treatment sample with quantified ROIs for comparison of solid tumour area and perinecrotic viral antigen positive tumour areas. Two subjects with pre-post-treatment pairs were examined (subjects 044 and 028). Scale bar = 50 μm. **(g)** Quantification of PD-L1 expression on total macrophage/microglial population in tumour near necrotic positive CAN-3110 region. (044: n = 3 ROIs (pre), 3 ROIs (Tumour), 6 ROIs (Necrotic); 028: n = 6 ROIs (pre), 3 ROIs (Tumour), 5 ROIs (Necrotic)) **(h)** post-treatment samples with CD163+ myeloid populations in both tumour and tumour-necrotic interface regions. Pre-treatment values are also shown. (044: n = 3 ROIs (pre), 3 ROIs (Tumour), 5 ROIs (Necrotic); 028: n = 6 ROIs (pre), 3 ROIs (Tumour), 5 ROIs (Necrotic)) **(i)** PD-L1 expression in CD163+ populations in perinecrotic interface regions. Pre-treatment values are also shown. DAPI blue nuclei enumeration, SOX2 white tumour nuclei enumeration, CD68 yellow pan-macrophage/microglia, CD163 orange macrophage/microglia. (044: n = 3 ROIs (pre), 3 ROIs (Tumour), 5 ROIs (Necrotic); 028: n = 6 ROIs (pre), 3 ROIs (Tumour), 5 ROIs (Necrotic)).

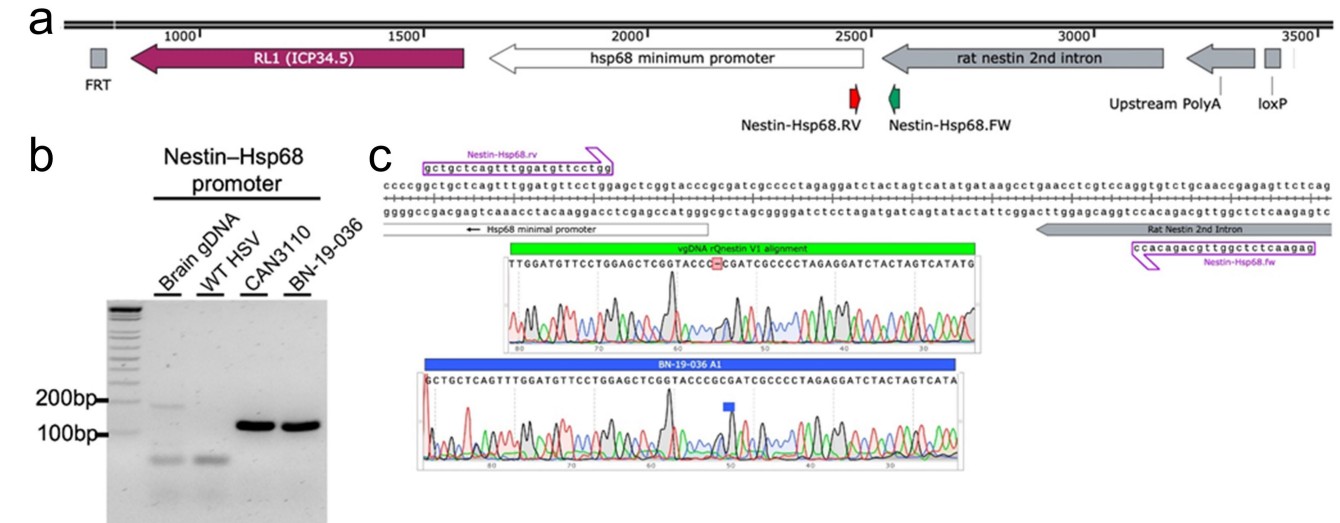

**Extended Data Fig. 8 | (Related to Sub-heading, *Persistence is linked to seronegativity*). Presence of CAN-3110 DNA in un-injected temporal lobe lesion of subject in Fig. 3b (8 months from injection in occipital lesion).** (a) Schematic of CAN-3110 viral genomic DNA showing the location of primers for Nestin-Hsp68 relative to other transcriptional cassette elements. (b) Gel electrophoresis of the PCR products from genomic DNA extracted from FFPE of postmortem brain from subject 014. Primers for Nestin-Hsp68 amplified a PCR product of ~112 bps within the promoter/enhancer transcriptional cassette. WT HSV represents a negative control with viral genomic DNA from HSV1 strain 17+. CAN-3110 represents a positive control with CAN-3110 viral genomic DNA amplifying a strong band of ~112 bp in the Nestin-Hsp68 promoter PCR reaction. BA-19-036 is the temporal lobe tumour that was not injected with CAN-3110 but was IHC-positive for HSV antigen (see Fig. 3b). The uncropped version of this gel is shown in Supplementary Fig. 1. Each PCR reaction was run in triplicate. (c) Sequencing reactions of CAN-3110 control and BA-19-036 PCR products show sequence homology with the original CAN-3110 sequence map.

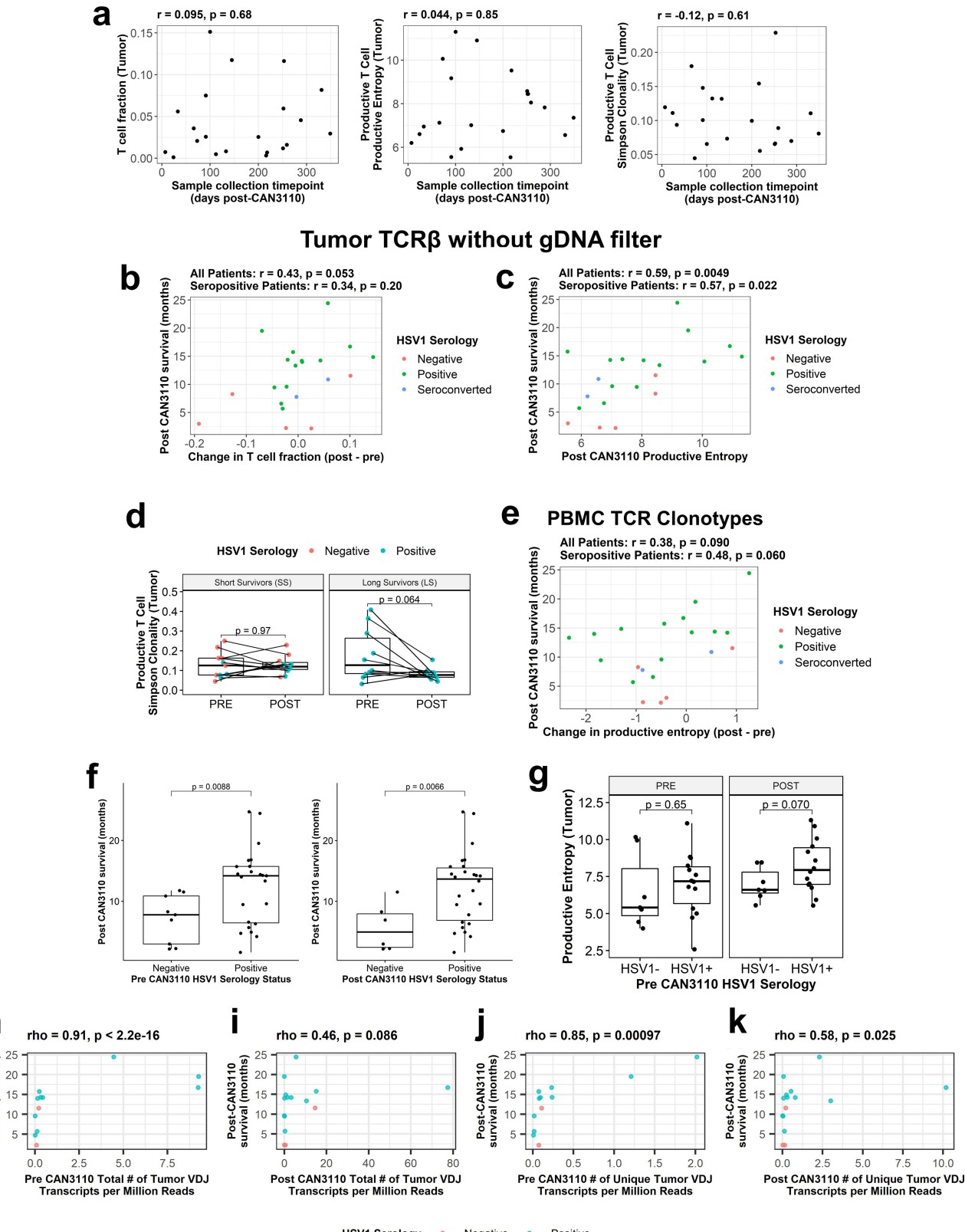

**Extended Data Fig. 9** | See next page for caption.

**Extended Data Fig. 9 | (Related to Sub-Heading, *T cell metrics are linked to survival*). TCR clonotype analyses for IDHwt rGBM patients. (a)** For the TCRβ DNAseq analysis of IDHwt patients (n = 21 interventions, 20 patients), Tumour T-cell fraction (left), productive T-cell entropy (middle), and productive T-cell Simpson Clonality (right) are not correlated with time from CAN-3110 treatment to tissue collection. **(b)** The change in tumour T cell fraction (Post minus Pre-CAN-3110) from all 21 interventions was analysed as a function of subject survival after CAN-3110 in all subjects and in the HSV seropositive ones. Unlike Fig. 4a, no gDNA filter was applied. **(c)** The post-CAN-3110 tumour productive entropy for the same 21 interventions is analysed as a function of subject survival after CAN-3110. Unlike Fig. 4b, no gDNA filter was applied. **(d)** Differences in Tumour Productive Simpson Clonality Indices in T cell TCRs from paired rGBM (n = 21 interventions, 20 patients) in the LS (> 1-year post-treatment survival) vs. SS (< 1-year post-treatment survival) patients as a function of CAN-3110 treatment. P value calculated using a Wilcoxon matched pairs signed rank test. **(e)** The change in productive entropy in TCRs from PBMCs for the same 21 tumour pairs is analysed as a function of subject survival after CAN-3110. Note that all panels omit patient 045 who had an early non-GBM mortality. **(f)** Comparison of post CAN-3110 survival time between pre- (left panel) or post- (right panel) CAN-3110 HSV1 serology positive (PRE: n = 23 interventions, 22 patients, POST: n = 26 interventions, 25 patients) and negative (PRE: n = 9 patients, POST: n = 6 patients) rGBM IDHwt patients for whom paired samples were available (e.g., for samples analysed in other panels in this figure). P values shown from 2-tailed Student's t-test. **(g)** Boxplot comparisons of tumour productive entropy values as grouped by timepoint (PRE or POST CAN-3110 treatment) and pre-CAN-3110 HSV1 serology status (negative or positive). P values calculated using two-tailed Student's t-test. Sample sizes for each group are as follows: 1) HSV1- POST = 7 patients; 2) HSV1 + POST = 14 interventions, 13 patients; 3) HSV1- PRE = 7 patients; 4) HSV1 + PRE = 14 interventions, 13 patients. Note that all panels omit patient 045 due to early non-GBM mortality. **(h-k)** Bulk RNAseq of tumours injected with CAN-3110. Both total **(h,i)** and unique **(j,k)** number of transcripts containing VDJ chain sequences were analysed before (n = 12 interventions, 11 patients) **(h, j)** or after (n = 15 interventions, 14 patients) **(i, k)** CAN-3110 and plotted against subject survival after CAN-3110. **(a-c and e)** Pearson's correlation coefficient r and p values (two-sided, based on t-distribution) are provided above each plot. **(h-k)** Spearman's correlation coefficient rho and p values (two-sided, based on t-distribution) are provided above each plot.

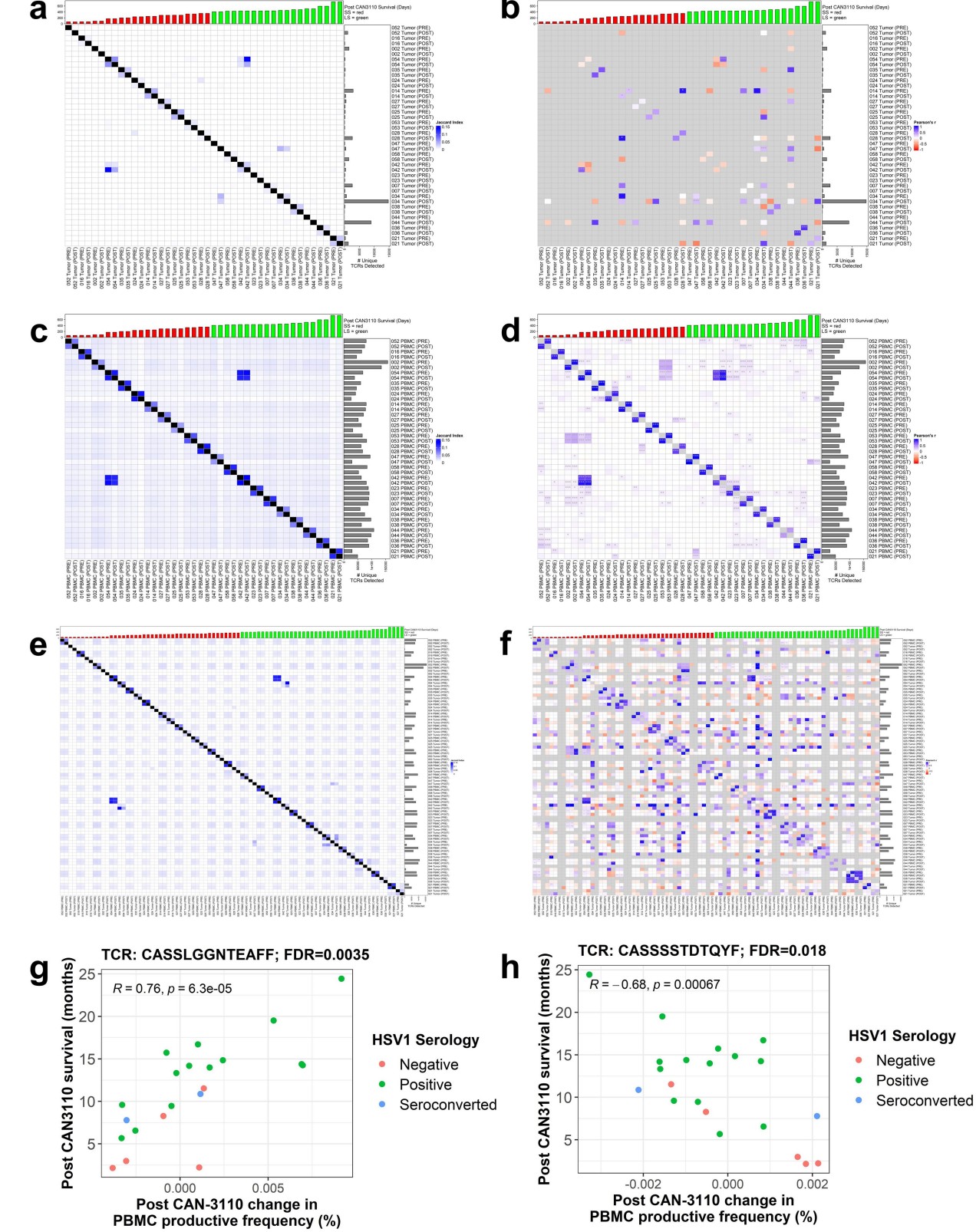

**Extended Data Fig. 10 |** See next page for caption.

**Extended Data Fig. 10 | (Related to *Specific public T cells are linked to survival*.) Public and Private TCR clonotypes in rGBM IDHwt (a, b), PBMCs (c, d) and combined (e, f).** Jaccard Indices heatmaps **(a, c, e)** and Pearson's correlation coefficient maps **(b, d, f)** are shown for amino acid-based shared (public) or unshared (private) TCRs between samples. For Jaccard maps **(a, c, e)**, colour in box provides a gradient with white indicating no shared TCRs and increasing shades of blue indicating more shared TCRs between tumours. For **(a-f)** the top row denotes survival for each subject with the red bars denoting the SS (short survivors defined as survival <1-year post CAN-3110) and green bars the LS (long survival defined as survival >1-year post CAN-3110) subjects and the small y axis showing survival days. The right Y and X axes denote each subject with paired Pre- and Post-CAN-3110 rGBM IDHwt, ordered based on respective overall survival time. The number of private TCRs is shown in the bar graphs to the right of the heatmaps. For **(b, d, f)**, each box represents a neutral (white), negative (red spectrum) or positive (blue spectrum) Pearson correlation coefficient between pre, post-CAN 3110 Tumour **(a, b)**, PBMCs **(c, d)**, or all combined **(e, f)**. In TILs, there were 5 TCRs publicly shared amongst 4 patients, 45 amongst 3 patients, and 792 between 2 patients. The remaining 41,756 tumour TCRs were private (i.e., not shared between patients). Grey boxes denote no shared TCRs between samples. Asterisks inside boxes denote significant Pearson's correlations p < 0.05, ** p < 0.01, *** p < 0.001 (2-sided, based on t-distribution). Note that 042 and 054 are the same individual treated at two different timepoints with CAN-3110. **(g, h)** PBMC TCRs for which post-CAN-3110 change in productive frequency associates with post-CAN-3110 survival (FDR ≤ 0.05 based on Pearson's correlation p values calculated 2-sided using t-distribution) in rGBM IDHwt with available TCRβ sequences (n = 21 interventions/20 patients). Patient 045 excluded due to early non-GBM related mortality. TCRβ sequences were included if present in PBMCs from 21 interventions with a median read count of at least 2 pre- or post-CAN-3110. Pearson's r, p (2-sided, based on t-distribution), and FDR are included in each plot. **(g)** CASSLGGNTEAFF[37,38] was detected in the TIL TCRs of two patients post CAN-3110. **(h)** CASSSSTDTQYF[39] was detected in the TIL TCRs of one patient pre CAN-3110.

**a**

| Patient | TCRβ Clonotype | Post CAN-3110 Survival (days) | Change in TCR Frequency (Tumor) | Change in TCR Frequency (PBMC) | Fisher's Exact FDR (Tumor) | Fisher's Exact FDR (PBMC) | TCRdb Search |
|---|---|---|---|---|---|---|---|
| 021 | CASSLFLAGAENEQFF | 744 | 7.06E-02 | 5.51E-08 | 6.81E-04 | 1.50E-09 | no exact matches |
| 021 | CASSQDRGDSPLHF | 744 | 4.50E-02 | 9.41E-05 | 5.25E-03 | 2.31E-103 | Found in 100 sample spanning tumor to healthy PBMCs. |
| 021 | CASTTPGGPDEQFF | 744 | 3.06E-02 | 1.16E-02 | 7.69E-05 | 2.49E-02 | no exact matches |
| 034 | CASSSSATSLEQFF | 452 | 3.28E-02 | 2.16E-11 | 8.78E-04 | 1.75E-06 | no exact matches |
| 023 | CASANAYEQYV | 434 | -1.37E-01 | 2.23E-04 | -8.34E-04 | 9.06E-04 | no exact matches |
| 036 | CASSLRYNTEAFF | 594 | -2.01E-01 | 3.77E-06 | -5.14E-03 | 4.50E-25 | Found in 21 samples spanning tumor to normal blood. |
| 036 | CATSEPSKNIQYF | 594 | -9.27E-02 | 9.61E-03 | -2.37E-03 | 1.73E-09 | no exact matches |
| 038 | CASSQDPGGQPQHF | 479 | -3.07E-01 | 1.89E-03 | -2.16E-02 | 0.00E+00 | Found in 13 samples spanning tumor to normal blood. |
| 042 | CASSVGRGFKNIQYF | 432 | -3.24E-02 | 1.44E-02 | -1.77E-02 | 0.00E+00 | no exact matches |

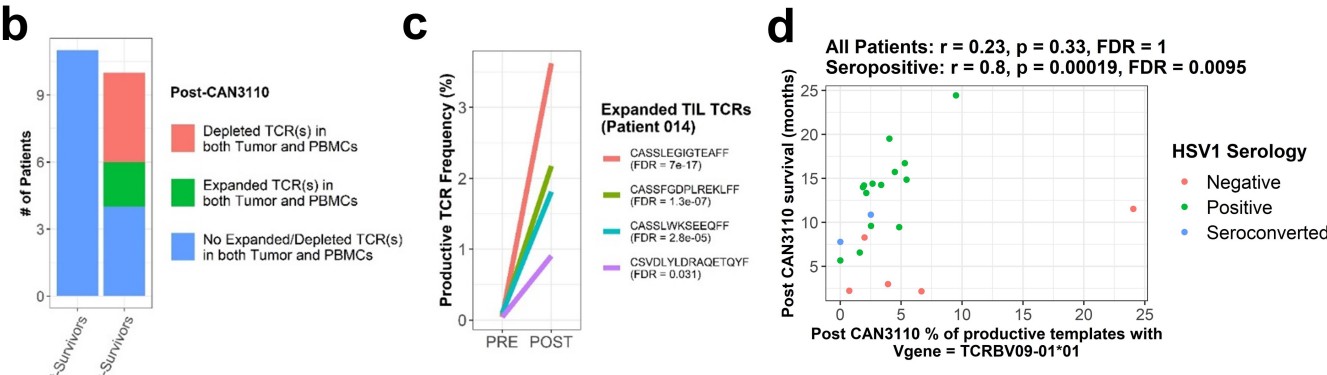

**Extended Data Fig. 11 | (Related to *Changes in T cell repertoire*). (a)** Table of TIL TCRs which were either statistically enriched post-CAN-3110 in both tumour and PBMCs or statistically depleted post-CAN-3110 in both TILs and PBMCs from the same patient for rGBM IDHwt patients with available TCRβ sequencing data. Statistical enrichment/depletion was determined via Fisher's Exact test with FDR correction on a per-patient basis. For TILs, FDR correction was applied across all detected TCRs. For PBMCs, FDR correction was applied only across TCRs that were statistically enriched/depleted in TILs and detected in PBMCs for that patient. The final column indicates whether the given TCR was reported in TCRdb (http://bioinfo.life.hust.edu.cn/TCRdb/#/) as of 11/04/2022. Further details of these TCR alterations are included in Supplementary Table 3. **(b)** Changes in Tumour and PBMC TCR repertoires were detected in long-survivors (post-CAN-3110 survival ≥ one-year post-CAN-3110) but not short-survivors (post-CAN-3110 survival < one-year post-CAN-3110) for rGBM IDHwt patients after CAN-3110. **(c)** TCR frequencies pre and post CAN-3110 for statistically expanded TIL TCRs (FDR ≤ 0.05) from patient 014, who was HSV1 seronegative both before and after CAN-3110 treatment (see also Fig. 3b and Extended Data Fig. 8). FDR values are calculated as in panel **a. (d)** Correlation between post-treatment TCRBV09-01*01 usage and post-treatment survival in IDHwt rGBM patients (n = 21 interventions, 20 patients) (excluding 045 due to early non-GBM mortality). Pearson's correlation coefficients r and p values (2-sided, based on t-distribution) are shown above the plot when calculated using all patients or using only HSV1 seropositive + seroconverted patients. FDR correction was performed using all V genes with a median usage > 0 in both pre- and post-treatment samples from these patients. GLM = Generalized Linear Model, FDR = False Discovery Rate.

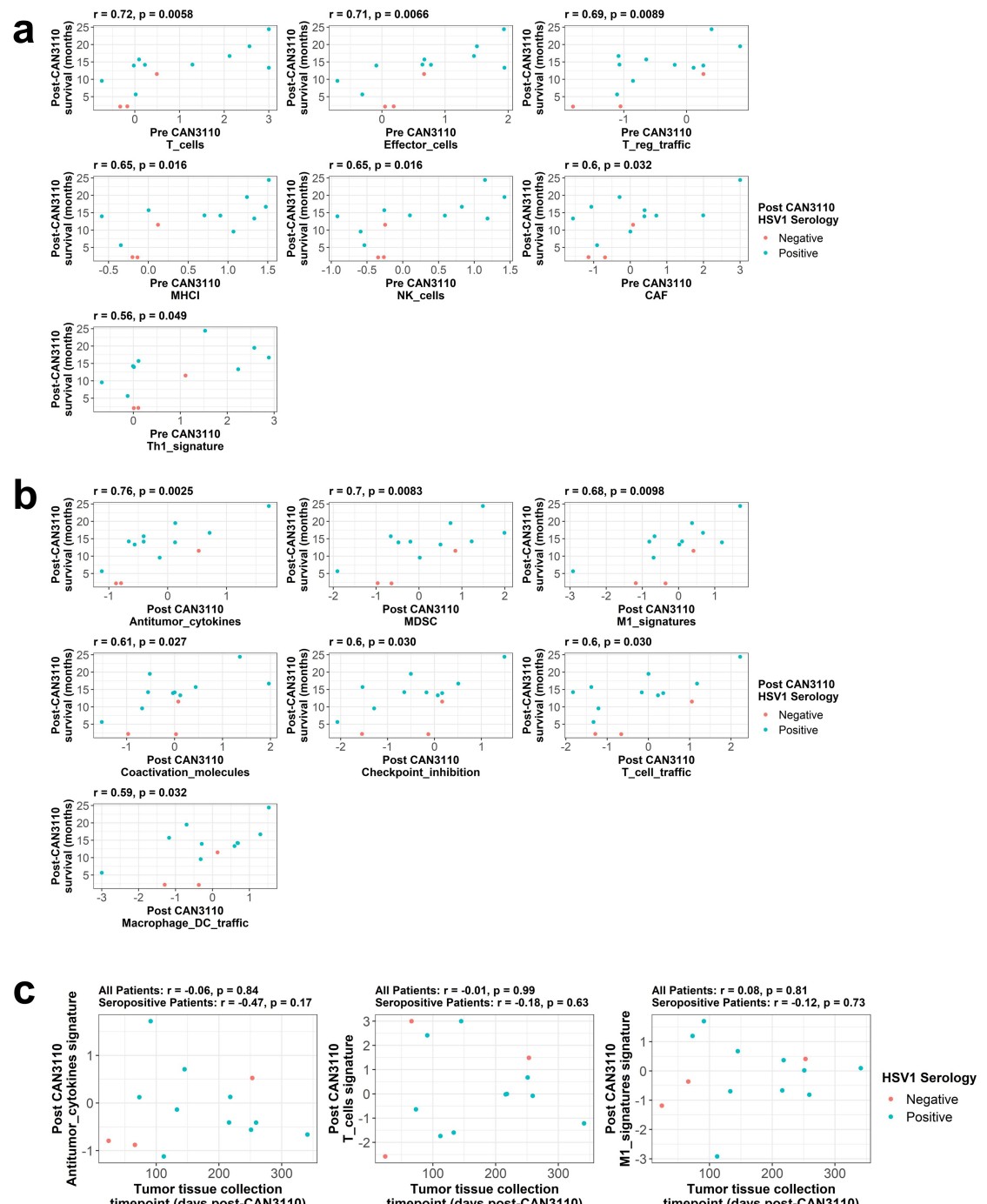

**Extended Data Fig. 12 | (Related to Sub-heading *Tumour immune signatures are linked to survival*). Few post-treatment RNAseq immune signatures are associated with survival when analysed using all available patients.** Bulk RNAseq immune signatures from paired pre- and post-treatment IDHwt rGBMs (n = 12 patients, 13 rGBMs) versus post CAN-3110 survival irrespective of HSV1 serology. **(a)** Pre-treatment Signatures which were significantly correlated (p ≤ 0.05, 2-sided, based on t-distribution) with post-treatment survival via Pearson's correlation when analysed using all available patients irrespective of

HSV1 serology. **(b)** Post-treatment Signatures which were significantly correlated (p ≤ 0.05, 2-sided, based on t-distribution) with post-treatment survival via Pearson's correlation when analysed using all available patients irrespective of HSV1 serology. Note that one patient (O45) was excluded from these analyses due to early non-GBM mortality. **(c)** Post-treatment Signatures were not significantly correlated (based on Pearson's r with 2-sided p values calculated via t-distribution) with time from CAN-3110 injection to post-treatment tissue collection.

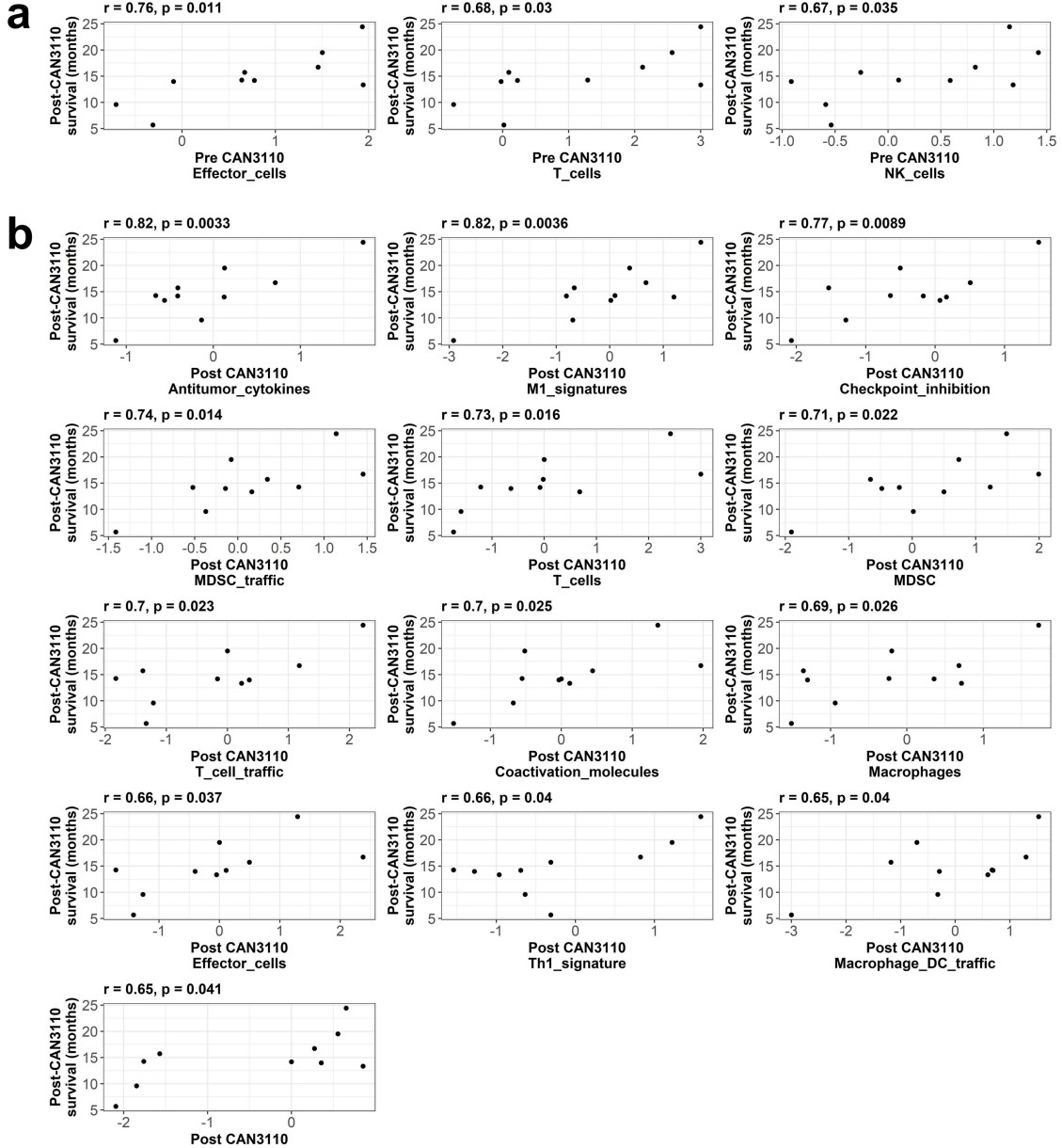

**Extended Data Fig. 13 | (Related to Sub-heading *Tumour immune signatures are linked to survival*). Many more post-treatment RNAseq immune signatures are associated with survival when analysed using only HSV1 seropositive patients than when analysed using all patients.** Bulk RNAseq immune signatures from paired pre- and post-treatment IDHwt rGBMs (n = 9 patients, 10 rGBMs) versus post CAN-3110 survival when analysed using only HSV1 seropositive patients. **(a)** Pre-treatment Signatures which were significantly correlated (p ≤ 0.05, 2-sided, based on t-distribution) with post-treatment survival via Pearson's correlation when analysed using all available patients irrespective of HSV1 serology. **(b)** Post-treatment Signatures which were significantly correlated (p ≤ 0.05, 2-sided, based on t-distribution) with post-treatment survival via Pearson's correlation when analysed using all available patients irrespective of HSV1 serology. Note that one patient (045) was excluded from these analyses due to early non-GBM mortality.

**Extended Data Table 1 | Demographics of subjects in dose-escalation phase 1 trial of CAN-3110 (Arm A: Cohorts 1-9)**

| | Number of Patients: 30* | |
|---|---|---|
| | N | % |
| AGE | | |
| Median Age | 56 | |
| Range | 30-74 | |
| SEX | | |
| Female | 13 | 43.3 % |
| Male | 17 | 56.7 % |
| KPS (BASELINE) | | |
| Median KPS | 90 | |
| MGMT | | |
| Methylated | 9 | 30.0 % |
| Unmethylated | 18 | 60.0 % |
| Partially Methylated | 3 | 10.0 % |
| IDH | | |
| Wild-Type | 25 | 83.3 % |
| Mutant | 5 | 16.7 % |
| Unknown | 0 | 0.0 % |
| TUMOR | | |
| Grade 3 | 4 | 13.3 % |
| Grade 4 | 26 | 86.7 % |
| Other | 0 | 0.0 % |
| | | |
| TUMOR VOLUME | | |
| Range Min.-Max ($mm^3$) | 357-92,041 | |
| Median ($mm^3$) | 11,577 | |
| SD ($mm^3$) | 22,802 | |
| SEM ($mm^3$) | 4,163 | |

(related to Sub-heading, Safety of CAN-3110 in rHGG/rGBM patients)

*One subject in cohort 9 (subject 042) was re-treated as part of cohort 10 (subject 054).

See explanation in Supplemental Text.

**Extended Data Table 2 | Demographics of subjects in dose-escalation phase 1 trial of CAN-3110 (Arm A: Cohort 10)**

| | Number of Patients: 12* | |
|---|---|---|
| | N | % |
| **AGE** | | |
| Median Age | 56 | |
| Range | 27-65 | |
| **SEX** | | |
| Female | 8 | 66.7 % |
| Male | 4 | 33.3 % |
| **KPS (BASELINE)** | | |
| Median KPS | 90 | |
| **MGMT** | | |
| Methylated | 6 | 50.0 % |
| Unmethylated | 6 | 50.0 % |
| Unknown | 0 | 0.0 % |
| **IDH** | | |
| Wild-Type | 8 | 66.7 % |
| Mutant | 4 | 33.3 % |
| Unknown | 0 | 0.0 % |
| **TUMOR** | | |
| Grade 3 | 3 | 25.0 % |
| Grade 4 | 9 | 75.0 % |
| Other | 0 | 0.0 % |
| | | |
| **TUMOR DIAMETER** | | |
| Range Min- Max ($mm^3$) | 1,549 – 23,749 | |
| Median ($mm^3$) | 9,804 | |
| Mean ($mm^3$) | 10,605 | |
| SD ($mm^3$) | 7,875 | |
| SEM ($mm^3$) | 2,274 | |

(related to Sub-heading, Safety of CAN-3110 in rHGG/rGBM patients)

*One subject in cohort 9 (subject 042) was re-treated as part of cohort 10 (subject 054).

See explanation in Supplemental Text.

**Extended Data Table 3 | Number of HGG recurrences at time of CAN-3110 accrual/HSV1 and HSV2 serology and blood biodistribution**

| Cohort | Patient ID | Recurrence | Pre-Injection HSV1 Serology | Post-Injection HSV1 Serology | Tumor HSV1 IHC After CAN-3110 | HSV1 Blood PCR Post-Injection | Pre-Injection HSV2 Serology | Post-Injection HSV2 Serology | HSV2 Blood PCR Post-Injection |
|---|---|---|---|---|---|---|---|---|---|
| 1 | 002 | 2nd | Negative | Negative | Pos | Not Detected | Negative | Negative | Not Detected |
| 1 | 005 | 3rd | Negative | N/A | Pos | Not Detected | Negative | N/A | Not Detected |
| 1 | 007 | 1st | Positive | Positive | Neg | Not Detected | Negative | Negative | Not Detected |
| 2 | 009 | 2nd | Positive | Positive | N/A | Not Detected | Negative | Negative | Not Detected |
| 2 | 011 | 2nd | Positive | Positive | N/A | Not Detected | Negative | Negative | Not Detected |
| 2 | 012 | 3rd | Positive | Positive | N/A | Not Detected | Positive | Positive | Not Detected |
| 3 | 006 | 1st | Positive | Positive | Neg | Not Detected | Negative | Negative | Not Detected |
| 3 | 014 | 2nd | Negative | Negative | Pos (multifocal†) | Not Detected | Negative | Negative | Not Detected |
| 3 | 015 | 1st | Positive | Positive | N/A | Not Detected | Negative | Negative | Not Detected |
| 4 | 016 | 1st | Negative | Negative | Pos (focal) | Not Detected | Positive | Positive | Not Detected |
| 4 | 017 | 1st | Positive | Positive | N/A | Not Detected | Negative | Negative | Not Detected |
| 4 | 019 | 2nd | Positive | Positive | Neg | Not Detected | Negative | Negative | Not Detected |
| 5 | 020 | 2nd | Positive | Positive | N/A | Not Detected | Negative | Negative | Not Detected |
| 5 | 022 | 1st | Positive | Positive | N/A | Not Detected | Negative | Negative | Not Detected |
| 5 | 021 | 2nd | Positive | Positive | Neg | Not Detected | Negative | Negative | Not Detected |
| 6 | 023 | 1st | Positive | Positive | Pos (weak) | Not Detected | Negative | Negative | Not Detected |
| 6 | 024 | 1st | Negative | Positive | Neg | Not Detected | Negative | Negative | Not Detected |
| 6 | 025 | 1st | Positive | Positive | Pos (focal) | Not Detected | Negative | Negative | Not Detected |
| 7 | 027 | 1st | Positive | Positive | Neg | Not Detected | Negative | Negative | Not Detected |
| 7 | 028 | 1st | Negative | Negative | Pos | Not Detected | Positive | Positive | Not Detected |
| 7 | 032 | 2nd | Positive | Positive | Pos (focal) | Not Detected | Negative | Negative | Not Detected |
| 8 | 033 | 3rd | Negative | Negative | Pos | Not Detected | Negative | Negative | Not Detected |
| 8 | 034 | 1st | Positive | Positive | Pos | Not Detected | Negative | Negative | Not Detected |
| 8 | 035 | 1st | Positive | Positive | Neg | Not Detected | Negative | Negative | Not Detected |
| 9 | 036 | 1st | Positive | Positive | Neg | Detected | Negative | Negative | Not Detected |
| 9 | 038 | 1st | Positive | Positive | Neg | Not Detected | Positive | Positive | Not Detected |
| 9 | 039 | 1st | Negative | Positive | N/A | Not Detected | Negative | Negative | Not Detected |
| 9 | 042* | 1st | Positive | Positive | Neg | Not Detected | Negative | Negative | Not Detected |
| 9 | 040 | 2nd | Positive | Positive | N/A | Not Detected | Negative | Negative | Not Detected |
| 9 | 044 | 1st | Positive | Positive | Pos | Not Detected | Negative | Negative | Not Detected |
| 10 | 045 | 1st | Negative | Negative | Pos | Not Detected | Negative | Negative | Not Detected |
| 10 | 046 | 1st | Positive | Positive | Neg | Not Detected | Positive | Positive | Not Detected |
| 10 | 047 | 3rd | Positive | Positive | Neg | Not Detected | Negative | Negative | Not Detected |
| 10 | 050 | 2nd | Negative | Negative | N/A | Not Detected | Negative | Negative | Not Detected |
| 10 | 052 | 1st | Negative | Negative | Neg | Not Detected | Positive | Positive | Not Detected |
| 10 | 048 | 4th | Positive | Positive | N/A | Not Detected | Negative | Negative | Not Detected |
| 10 | 049 | 1st | Positive | Positive | N/A | Not Detected | Positive | Positive | Not Detected |
| 10 | 051 | 4th | Negative | Positive | Neg | Not Detected | Negative | Negative | Not Detected |
| 10 | 053 | 4th | Negative | Positive | Pos | Not Detected | Negative | Negative | Not Detected |
| 10 | 054* | 2nd | Positive | Positive | Neg | Not Detected | Negative | Negative | Not Detected |
| 10 | 057 | 1st | Negative | Negative | N/A | Not Detected | Positive | Positive | Not Detected |
| 10 | 058 | 2nd | Positive | Positive | Neg | Not Detected | Negative | Negative | Not Detected |

(related to Sub-headings, Safety of CAN-3110 in rHGG/rGBM patients and to Sub-Heading, HSV1 serology predicts efficacy)

*One subject in cohort 9 (subject 042) was re-treated as part of cohort 10 (subject 054). See explanation in Supplemental Text. †Multifocal staining pattern could be caused by tissue fragmentation.

**Extended Data Table 4 | All treatment phase AEs (grade 1 or 2) reported to date possibly, likely or definitely related to CAN-3110**

| Study ID | Dose Cohort | Time since injection (days) | Category | Adverse Events | CTC Grade | Relation to CAN-3110 |
|---|---|---|---|---|---|---|
| 017 | Arm A 3x10$^7$ | 28 | Musculoskeletal and connective tissue disorders | Muscle Weakness Lower Limb | 1 | Possible |
| | | | General disorders and administration site conditions | Fatigue | 1 | Possible |
| 017 | Arm A 3x10$^7$ | 149 | Nervous System Disorders | Edema Cerebral | 1 | Possible |
| 032 | Arm A 1x10$^9$ | 13 | Investigations | Alanine Aminotransferase Increased | 1 | Possible |
| | | | Blood and Lymphatic System | Low Eosinophil Count | 1 | Possible |
| 035 | Arm A 3x10$^9$ | 0 | Nervous System Disorders | Expressive Aphasia | 1 | Possible |
| 035 | Arm A 3x10$^9$ | 1 | Investigations | Platelet Count Decreased | 1 | Possible |
| | | | Investigations | Lymphocyte Count Decreased | 1 | Possible |
| 036 | Arm A 1x10$^{10}$ | 152 | Nervous System Disorders | Right Arm Joint Position Sense Loss | 1 | Possible |
| 039 | Arm A 1x10$^{10}$ | 9 | Nervous System Disorders | Edema Cerebral | 1 | Possible |
| | | | Nervous System Disorders | Headache | 2 | Possible |
| | | | Nervous System Disorders | Speech | 2 | Possible |
| 046 | Arm A 1x10$^9$ (2 ml) | 1 | General disorders and administration site conditions | Fever | 1 | Possible |
| 047 | Arm A 1x10$^9$ (2 ml) | 1 | General disorders and administration site conditions | Fever | 1 | Possible |
| 052 | Arm A 1x10$^9$ (2 ml) | 1 | General disorders and administration site conditions | Fever | 1 | Possible |
| 053 | Arm A 1x10$^9$ (3 ml) | 19 | Nervous System Disorders | Seizure | 2 | Possible |
| 057 | Arm A 1x10$^9$ (2 ml) | 2 | Musculoskeletal and connective tissue disorders | Muscle Weakness Upper Limb | 1 | Possible |

(related to Sub-heading, Safety of CAN-3110 in rHGG/rGBM patients).
All grade 3s are reported in Table 1.

# Reporting Summary

## Statistics

For all statistical analyses, confirm that the following items are present in the figure legend, table legend, main text, or Methods section.

| n/a | Confirmed | |
|---|---|---|
| ☐ | ☒ | The exact sample size (*n*) for each experimental group/condition, given as a discrete number and unit of measurement |
| ☐ | ☒ | A statement on whether measurements were taken from distinct samples or whether the same sample was measured repeatedly |
| ☐ | ☒ | The statistical test(s) used AND whether they are one- or two-sided<br>*Only common tests should be described solely by name; describe more complex techniques in the Methods section.* |
| ☐ | ☒ | A description of all covariates tested |
| ☐ | ☒ | A description of any assumptions or corrections, such as tests of normality and adjustment for multiple comparisons |
| ☐ | ☒ | A full description of the statistical parameters including central tendency (e.g. means) or other basic estimates (e.g. regression coefficient) AND variation (e.g. standard deviation) or associated estimates of uncertainty (e.g. confidence intervals) |
| ☐ | ☒ | For null hypothesis testing, the test statistic (e.g. *F*, *t*, *r*) with confidence intervals, effect sizes, degrees of freedom and *P* value noted<br>*Give P values as exact values whenever suitable.* |
| ☒ | ☐ | For Bayesian analysis, information on the choice of priors and Markov chain Monte Carlo settings |
| ☒ | ☐ | For hierarchical and complex designs, identification of the appropriate level for tests and full reporting of outcomes |
| ☐ | ☒ | Estimates of effect sizes (e.g. Cohen's *d*, Pearson's *r*), indicating how they were calculated |

*Our web collection on statistics for biologists contains articles on many of the points above.*

## Software and code

Policy information about availability of computer code

| Data collection | Medical record data was collected using EPIC (v May2022) and InForm (v2.3). Immunofluorescence image acquisition was performed using the Mantra multispectral imaging platform (Vectra 3, PerkinElmer) |
|---|---|
| Data analysis | Cell identification for multiplexed immunofluorescence was performed using Akoya Inform Automated Image Analysis Software version 2.4.8. Oncoprint genomic profiling was analyzed using R 4.2.1, RStudio 2022.07.2+576, and the Oncoprint function of the ComplexHeatmap 2.12.1 package. Data plotting and statistical analyses were performed using R (version 4.1.0) and RStudio (version 2022.2.3.492) along with the following packages: openxlsx (version 4.2.5); ggplot2 (version 2_3.3.5); tidyverse (version 1.3.1); rstatix (version 0.7.0); ggpubr (version 0.4.0); survival (version 3.2-11); gridExtra (version 2.3); survminer (version 0.4.9); doSNOW (version 1.0.19); foreach (version 1.5.1); ComplexHeatmap (version 2.8.0); and RColorBrewer (version 1.1-2). RNA-seq reads were aligned using Kallisto v0.42.4. ssGSEA algorithm was used to calculate gene signature scores (https://doi.org/10.1016/j.cell.2021.04.014). MIXCR v.3.0.13 was used to analyze T and B cell receptor repertoire from the RNA-seq samples. MRI segmentations were performed manually using 3D Slicer for cohorts 1-9 or using SmartBrush Software (version 3.0.0.92, BrainLab AG, Munich, Germany) for cohort 10. |

For manuscripts utilizing custom algorithms or software that are central to the research but not yet described in published literature, software must be made available to editors and reviewers. We strongly encourage code deposition in a community repository (e.g. GitHub). See the Nature Portfolio guidelines for submitting code & software for further information.

## Data

Policy information about availability of data

All manuscripts must include a data availability statement. This statement should provide the following information, where applicable:
- Accession codes, unique identifiers, or web links for publicly available datasets
- A description of any restrictions on data availability
- For clinical datasets or third party data, please ensure that the statement adheres to our policy

Patient responses, demographic information, and safety outcomes, as well IHC quantifications and RNAseq gene signature scores are available within the paper and its Supplementary Information. Raw RNA sequencing and TCRβ DNA sequencing files have been deposited in a controlled access repository at the database of Genotypes and Phenotypes (dbGaP): http://www.ncbi.nlm.nih.gov/projects/gap/cgi-bin/study.cgi?study_id=phs003378.v1.p1

## Human research participants

Policy information about studies involving human research participants and Sex and Gender in Research.

| Reporting on sex and gender | Biological gender was included in a CoxPH multivariate analysis of post-treatment survival alongside other potential covariates of survival. Gender was not determined to be a significant factor in patient survival following therapy in this trial, and, given the small sample size available in a phase I trial, no further analyses of gender were performed. |
| --- | --- |
| Population characteristics | Population characteristics are fully described in Extended Data Supplementary Tables 1A-1C of the manuscript. |
| Recruitment | Potentially eligible subjects were recruited from: 1- subjects seen or referred to our brain tumor clinics at Dana-Farber Cancer Institute and Brigham and Women's Hospital, 2- subjects made aware of the study via the clinicaltrials.gov website, 3- subjects referred from national patient referral organizations such as the National Brain Tumor Consortium, 4- subjects referred by direct physician or other healthcare professional, 5- subjects made aware via word of mouth, or via personal searches. Bias in patient selection is always possible. Potentially eligible patients were accrued by internal referral, by external referrals, by patients seeking care after finding out about the trial or by patient care clinical trial networks referring patients for consideration into the trial. To minimize bias, an independent neurosurgeon, external to our institution (Dr. Ekkehard Kasper, St. Elizabeth's Medical Center, Boston MA) reviewed eligibility for each patient's MRIs, history, medical exams before proceeding with the trial. |
| Ethics oversight | This phase 1 clinical trial was reviewed and approved by NIH RAC Office of Biotechnology Affairs (NIH no 1104-1100) and the IRB from the DFCI (no 16-557). The IND Sponsor was Dr. Chiocca (IND 16380). |

Note that full information on the approval of the study protocol must also be provided in the manuscript.

# Field-specific reporting

Please select the one below that is the best fit for your research. If you are not sure, read the appropriate sections before making your selection.

☒ Life sciences          ☐ Behavioural & social sciences          ☐ Ecological, evolutionary & environmental sciences

For a reference copy of the document with all sections, see nature.com/documents/nr-reporting-summary-flat.pdf

# Life sciences study design

All studies must disclose on these points even when the disclosure is negative.

| Sample size | Since this was a 3+3 dose escalation phase I trial, sample sizes for cohorts 1-9 were defined by the dose-escalation schema and were not sized to obtain statistical power for correlative analyses. As such, no statistical methods were used to pre-determine sample sizes. Sample size for cohort 10 was based on feasibility as a small exploratory expansion cohort, again, with no statistical power analyses being performed when selecting sample size in this phase I trial. In this exploratory analysis, samples sizes for immunohistochemistry, TCRbeta sequencing, and RNAsequencing were dictated by the availability of high-quality tissues for staining/DNA or RNA extraction. |
| --- | --- |
| Data exclusions | As clearly stated in all relevant analyses, patient 045 was excluded from analyses due to having experienced a non-GBM mortality shortly after trial enrollment. This exclusion criteria was not established prior to trial enrollment; however, most analyses were never performed in a way that included patient 045, and we are unaware of any analysis which would have resulted in a different outcome had patient 045 been included. |
| Replication | Due to the expense and time involved with conducting clinical trial research, replication was not feasible for any of the experiments presented in this manuscript. It is our intent to replicate findings from this manuscript in later clinical trials and separate manuscripts as the data becomes available over the coming years. |
| Randomization | As routinely done for Phase 1 studies, randomization was not possible in this study, because only one cohort was open for recruitment at any given time. |

| Blinding | Blinding was not possible for patients caregivers in this study, because, given that this was a dose-escalation trial phase 1 study , only one cohort was open for recruitment at any given time. This fact also made it impractical to blind researchers to patient group when determining tumor volumes or immune infiltration. However, rigorous and consistent criteria were applied when grading patients (as described in supplemental methods), and we do not believe significant bias occurred due to this lack of blinding. This is especially true since we see concordance in the data between potentially subjective metrics of immune infiltration (i.e. pathological quantifications) and strictly quantitative metrics (i.e. ImmunoSeq quantifications). |
|---|---|

# Reporting for specific materials, systems and methods

We require information from authors about some types of materials, experimental systems and methods used in many studies. Here, indicate whether each material, system or method listed is relevant to your study. If you are not sure if a list item applies to your research, read the appropriate section before selecting a response.

## Materials & experimental systems

| n/a | Involved in the study |
|---|---|
| ☐ | ☒ Antibodies |
| ☒ | ☐ Eukaryotic cell lines |
| ☒ | ☐ Palaeontology and archaeology |
| ☒ | ☐ Animals and other organisms |
| ☐ | ☒ Clinical data |
| ☒ | ☐ Dual use research of concern |

## Methods

| n/a | Involved in the study |
|---|---|
| ☒ | ☐ ChIP-seq |
| ☒ | ☐ Flow cytometry |
| ☐ | ☒ MRI-based neuroimaging |

## Antibodies

| Antibodies used | HSV-1 polyclonal antibody (Dako Polyclonal); CD4 (Dako 4B12), CD8 (Dako 144D), CD20 (Dako L26), Nestin (Cell Signaling Technologies 10C2), Nectin-1/CD111 (Santa Cruz Biotech CK6), Sox2 (Cell Signaling Technologies D6D9), CD68 (Dako PG-M1), CD163 (Novocastra 10D6), PD-L1 (Cell Signaling Technologies E1L3N). Antibody dilutions can be found it the supplemental methods. |
|---|---|
| Validation | For multiplexed immunofluorecence, all of the antibodies are commonly used. Each antibody was first optimized by standard IHC to confirm fidelity of the staining, then adapted to single-immunofluorescence staining before combining antibodies together into a multiplex immunofluorescence panel. In single-immunofluorescence, repeated rounds of optimization include testing different antigen retrieval conditions, diluents and a wide range of antibody concentrations. In multiplex, different panel conditions are tested to ensure high signal to noise for each individual marker, while eliminating bleedthrough, crosstalk between channels, and nonspecific staining.<br>For chromogenic immunohistochemistry, all staining was perfomed utilizing commercially available antibodies optimized for staining formalin-fixed paraffin-embedded tissue sections, with staining performed in a CAP certified laboratory. |

## Clinical data

Policy information about clinical studies

All manuscripts should comply with the ICMJE guidelines for publication of clinical research and a completed CONSORT checklist must be included with all submissions.

| Clinical trial registration | NCT03152318 |
|---|---|
| Study protocol | Study protocol can be accessed at: https://www.dropbox.com/s/rj035h42svm7i71/16-557%20Protocol%20Cohort%2010%20Arm%20A%2003FEB2020%20-%20clean.pdf?dl=0 |
| Data collection | All data were collected at the Brigham and Women's Hospital and/or Dana Farber Cancer Institute. In some cases for patients who were not local, some data was collected at their outside hospital and physician place of care. Outside collections occurred for some patients who lived in Florida, New York, New Hampshire, Maine, Vermont, Connecticut, Rhode Island. Period times for recruitment were from September 2017 until December 2020. Data collection occurred between September 2017 until March 2023 when a subject of arm A underwent resection of recurrence of their high grade glioma after treatment with CAN-3110. |
| Outcomes | Outcome descriptions are too lengthy to include here, but are described in detail in the supplemental methods sections:<br>1. Clinical protocol: definition of adverse event (AE), serious adverse event (SAE), dose limiting toxicity (DLT), and maximum tolerated dose (MTD).<br>2. Clinical Protocol: response assessment |

## Magnetic resonance imaging

### Experimental design

| Design type | Standard MRI sequences for brain tumors (with and without gadolinium) obtained preoperatively, withing 72 hours post-operatively and then every 8 weeks. In most cases additional MRIs were available before and after intrevention |
|---|---|

| | |
|---|---|
| Design specifications | Standard specifications for routine clinical MRIs. |
| Behavioral performance measures | Behavioral performance measures were not a part of this study. |

## Acquisition

| | |
|---|---|
| Imaging type(s) | Regular MRI sequences for all, including perfusion imaging. If clinically indicated (close to eloquent brain), funtional imaging with tractography performed too |
| Field strength | 3 Tesla |
| Sequence & imaging parameters | T1, T2, FLAIR, T1 with gadolinium, DWI, ADC, DTI |
| Area of acquisition | Whole brain |

Diffusion MRI ☐ Used ☒ Not used

## Preprocessing

| | |
|---|---|
| Preprocessing software | Segmentations were performed manually by a trained neurosurgeon using 3D Slicer (v5.1.0-2022-10-31 or previous) for cohorts 1-9. or using SmartBrush Software (version 3.0.0.92, BrainLab AG, Munich, Germany) for cohort 10. |
| Normalization | No pre-processing was performed prior to segmentation. |
| Normalization template | No pre-processing was performed prior to segmentation. |
| Noise and artifact removal | No pre-processing was performed prior to segmentation. |
| Volume censoring | No pre-processing was performed prior to segmentation. |

## Statistical modeling & inference

| | |
|---|---|
| Model type and settings | No statistical modeling/inference was performed using the MRI data. |
| Effect(s) tested | No statistical modeling/inference was performed using the MRI data. |

Specify type of analysis: ☐ Whole brain ☒ ROI-based ☐ Both

Anatomical location(s) | Volumes were obtained specifically for tumor regions of the brain. |

| | |
|---|---|
| Statistic type for inference (See Eklund et al. 2016) | No statistical modeling/inference was performed using the MRI data. |
| Correction | No statistical modeling/inference was performed using the MRI data. |

## Models & analysis

| n/a | Involved in the study |
|---|---|
| ☒ | ☐ Functional and/or effective connectivity |
| ☒ | ☐ Graph analysis |
| ☒ | ☐ Multivariate modeling or predictive analysis |

