## [Peer Review File · Nature]

Manuscript Title: Oncolytic HSV1 clinical trial links immunoactivation to survival in glioblastoma

Redactions – unpublished data

Reviewer Comments & Author Rebuttals

Reviewer Reports on the Initial Version:

Referees' comments:

Referee #1 (Remarks to the Author):

The manuscript by Ling, Solomon et al describes a phase 1/ 2 clinical trial testing a novel oncolytic herpes simplex virus CAN-3110 for treatment of malignant high grade glioma. Although immunotherapy using oncolytic viruses in this patient group has been described previously using other oncolytic viruses, this is the first study to describe the effects of treatment on development of an expanded T cell repertoire and relating these changes to patient survival. Moreover, discussions in the field regarding the role of preexisting antibodies in OV therapy have thus far not been addressed clinically on such a large scale. The study also presents extensive immune-, tumor- and virus-associated analyses of tissues derived from treated patients. The work is novel and brings significant advancement to the field of oncolytic virus therapy for this deadly disease.

General points:

- Supplemental figure 5, Swimmers plots, shows that 2 of 3 patients surviving beyond 24 months were treated with checkpoint inhibitors (nivo and pembro) after the oncolytic virus treatment. This interesting finding can be mentioned and reflected upon.
- Why do IDH mut astrocytomas have a much worse response compared to IDHwt, in the light of the fact that IDH mut tumors are regarded as more immunogenic.
- Patient nr 19 had a nestin negative tumor at time of CAN-3110 treatment yet had very long survival of 40 months, what could be the explanation?
- PCR analysis in supplemental figure 10 shows viral spreading to the other tumor site, but to what extent does it demonstrate 'likely ongoing replication', as mentioned in text, at that site? Maybe just prolonged presence in latent form?
- Is there any relation between (late) extent or amount of intratumoral virus presence and response to treatment?
- Figure 4: Is it possible to extract any information on the phenotype of T cells, eg are the T cells with enriched clones exhausted? memory? effector? naive?
- Supplemental fig 13 shows the main checkpoint inhibitors such as PD1 and CTLA4, however other markers, which are considered hallmarks of intratumoral tumor-reactive CD8 T cells are missing, such as Tim-3, CD38, TCF-1 CD103, CD39 (among others).
- For the assessment of TCR diversity it could also be of interest to determine CDR3 length of TILs between LS and SS

Minor comments:

- Why not also show HSV preexisting antibodies in IDH mutant patients (n=9 in total) to see if same trend is found?
- Fig 2C peri-necrotic core staining of CD8 and CD4 would benefit from insertion of selected area enlargements at higher magnification to show specific staining.
- Fig 2D Mention type of cell (CD4, CD8, CD20) above the graphs for easier reading
- Fig 2E what about pre peri-necrotic?
- Present CD4, 8, and 20 in same order each figure for clarity
- Add to X axis title of 3A and 3B 'in tumor' and 'in PBMCs' for easier graph reading
- Graph 6B is redundant, the information is already provided in 6A, authors can just mention p values for subgroups in text.
- Extended data file page 12, replace brain tissue collection by brain tumor tissue collection

Referee #4 (Remarks to the Author):

This manuscript describes an innovative and a well conducted first in human phase I trial of CAN-3110, an oncolytic herpes virus strain that expresses the viral neurovirulence ICP34.5 gene under the transcriptional control of the Nestin promoter in patients with recurrent GBM. In addition, the manuscript presents a before and after treatment comprehensive correlative analysis of tumor and peripheral blood including serologic, immunohistochemistry as well as RNA and T cell receptor DNA sequence analysis. Overall, the findings provide novel insights into the mechanism of action of the CAN-3110 that have broad translational implications for the oncolytic virotherapy field.

Comments to Authors:

- The investigators have obtained post treatment tumor tissue at different time points. It would be important to know what other antitumor treatments the patients have received prior to the surgical sampling being performed since this could have impacted the correlative analysis results.
- For patients in whom surgical sampling on more than one occasions was performed, it would be desirable to see individual patient trends over time regarding T cell infiltrates, RNA and TCR DNA

sequence analysis.

- The investigators comment that in 12/29 tumors HSV antigen was detected by IHC even several months after CAN-3110 injection. PCR data confirming the presence of CAN-3110 specific viral RNA was only presented for one patient, however. Was PCR performed in all patients with HSV positivity in IHC and what were the respective findings?
- Increased TCR β diversity was noted. It would be important to know if this reflects antitumor or antiviral immune response.
- What is the definition of long-term and short-term survivors for the data presented in Supplementary Figure 13C and D? It is possible that other treatments that these patients have received after progression on study could have impacted these results that they cannot solely be attributed to the viral treatment.
- Page 8: “We found that post-treatment survival was significantly longer for patients who were serologically HSV+ pre- or post-treatment with CAN-3110.” Please include median OS and range for the two groups in addition to p-values.

Author Rebuttals to Initial Comments:

POINT BY POINT RESPONSE TO REFEREES

Referees' comments:

Referee #1 (Remarks to the Author):

The manuscript by Ling, Solomon et al describes a phase 1/2 clinical trial testing a novel oncolytic herpes simplex virus CAN-3110 for treatment of malignant high grade glioma. Although immunotherapy using oncolytic viruses in this patient group has been described previously using other oncolytic viruses, this is the first study to describe the effects of treatment on development of an expanded T cell repertoire and relating these changes to patient survival. Moreover, discussions in the field regarding the role of preexisting antibodies in OV therapy have thus far not been addressed clinically on such a large scale. The study also presents extensive immune-, tumor- and virus-associated analyses of tissues derived from treated patients. The work is novel and brings significant advancement to the field of oncolytic virus therapy for this deadly disease.

General points:

-Supplemental figure 5, Swimmers plots, shows that 2 of 3 patients surviving beyond 24 months were treated with checkpoint inhibitors (nivo and pembro) after the oncolytic virus treatment. This interesting finding can be mentioned and reflected upon.

Answer: Based on this excellent comment, we have added the following sentences to the discussion:

“Taken together, positive HSV1 serology with the observed changes in T cell clonotypes, including public ones, suggest a more efficacious immune response, characterizing subjects whose immune system is more “fit” and can mount a more effective antiviral and possibly antitumor adaptive response. It is notable that 2 of the longest survivors were treated with immune checkpoint inhibition (ICI) after their injected tumors were resected (see subject 019 and 021 in Extended Data Figures 3C-D), based on the post-injection finding of extensive TILs. We could speculate that CAN-3110 mediated conversion of the TME into an inflamed one might have rendered ICI therapy more effective.”

- Why do IDH mut astrocytomas have a much worse response compared to IDHwt, in the light of the fact that IDH mut tumors are regarded as more immunogenic.

Answer: We thank the reviewer for this comment. Interestingly, 2 of the 4 IDHmut astrocytomas with the shortest survival (005 and 033 with OS of about 5 and 3 months, respectively) were both HSV1 seronegative and positive for HSV antigen by IHC, while one of 4 (046) instead with OS of about 22 months was HSV1 seropositive and positive for HSV antigen by IHC. For the last patient (050) we did not have post-treatment tissue. Therefore, the poor survival of this group is consistent with our findings with the GBM IDHwt group. It should also be noted that, despite the large apparent difference in survival between IDHmut astrocytomas and IDHwt rGBMs in Figure 1B, this difference was not statistically significant (likelihood ratio $p = 0.8$) due to the small number of IDHmut astrocytomas in our study.

To address the excellent comment of the reviewer, we have added this new text in the results section:

Similarly, the trend towards longer survival for HSV1 seropositive patients was observed in the IDH mutant rAA patients (Extended Data Figure 5B), though the sample size for this group is too small to draw definitive conclusions.

And also added new Extended Data Figure 5B with legend:

Extended Data Figure 5 (related to Sub-Heading, *Positive HSV1 serology as an independent predictor of CAN-3110 efficacy in rGBM subjects*)

(A) Comparative analyses of differential survival based on HSV1 pre-operative (left panel) and post-operative (right panel) serology for the 31 rGBM IDHwt subjects (32 interventions). P value from two-tailed Student's t-test. **(B)** Kaplan-Meier survival curves for IDH mutant patients as divided by pre-treatment HSV1 serological status. Median OS months with 95% confidence intervals = rAA HSV1-: 3.5 [2.6 - Inf]; rAA HSV1+: 22.5 [-Inf - Inf]; rAO HSV1-: not reached; rAO HSV1+: 39.9 [20.1 - Inf].

-Patient nr 19 had a nestin negative tumor at time of CAN-3110 treatment yet had very long survival of 40 months, what could be the explanation?

Answer: This subject was an IDHmut, 1p/19q co-deleted anaplastic oligodendroglioma, whose biology is known to portend long survival. The subject also had significant changes in the TME after injection with high level of immune activating transcripts and conversion of M2 to M1 myeloid signatures (data not shown, but this data from this patient was presented orally at ASCGT 2021, ASO2021 and SNO2021...powerpoint slides are available, if reviewer requests them). These factors likely were more relevant than robust viral replication mediated by nestin expression.

-PCR analysis in supplemental figure 10 shows viral spreading to the other tumor site, but to what extent does it demonstrate 'likely ongoing replication', as mentioned in text, at that site? Maybe just prolonged presence in latent form?

Answer: We do not think that this would be latent virus since viral antigens would not be expressed during latency. Since it was post-mortem tissue, we could not extract RNA of quality sufficient enough for RT-PCR of viral transcripts. We believe that there was "likely" ongoing replication based on the amount of time since injection (8 months), the fact this was a non-injected tumor (the injected tumor was in occipital lobe while the IHC positive CAN-3110 was in the temporal lobe, and the strength of the IHC positivity coupled with extensive

CD8+ T cells around it (as shown in figure 3B). Please note that in order to render extended data more concise, this figure is now Extended Data Figure 8.

-Is there any relation between (late) extent or amount of intratumoral virus presence and response to treatment?

Answer: We thank the reviewer for this excellent comment. Based on this, we have extensively modified the paper in several areas to show that intratumoral viral presence actually correlates with worst outcome and HSV1 seronegativity, suggesting that failure to clear virus may be a marker of a less effective immune response.

In the summary we now state:

“Surprisingly, positive HSV1 serology was significantly associated with both improved survival and clearance of CAN-3110 from injected tumors.”

In the main result/ text section, we changed a sub-heading to state:

Prolonged persistence of CAN-3110 in injected tumor associates with negative HSV1 serology.

In this section, we then added the following new text

“We then asked if the prolonged persistence of CAN-3110 in injected tumors was associated with HSV1 serological status. Significantly, we discovered that oHSV persistence indeed significantly correlated with absence of HSV1 seropositivity either before or after CAN-3110 (Figure 3C-D). These findings thus suggested that oHSV persistence in injected rHGGs/rGBMs may have been due to absence of a robust anti-HSV1 immune response. Coupled with the extended survival for subjects with positive HSV1 serology (Figure 1C), this suggests that tumor clearance of CAN-3110 characterized subjects with an improved survival response to CAN-3110.”

We added new panels C and D with corresponding Legend to Figure 3:

Figure 3 – Ling et al.

Figure 3 - CAN-3110 persistence in injected rHGG/rGBM associates with negative HSV1 serological status either before or after therapy. (A) Positive oHSV IHC from 2 subjects. The top 3 panels show MRIs before and 41 days after CAN-3110 injection (10^6 PFUs) from subject 005. Positive oHSV IHC was visualized in the large area of tumor necrosis. The area was also positive for oHSV DNA by PCR and ICP22 oHSV transcript by RT-PCR (**data not shown**). The bottom 3 panels show MRIs from subject 028 before and 253 days after CAN-3110 injection (10^9 PFUs). Positive oHSV IHC was visualized in the area of resected tumor necrosis as well as positive ICP22 oHSV transcript by RT-PCR (**data not shown**). **(B)** Subject 014 had multifocal GBMs in the left temporal and left occipital lobes. The left occipital lobe lesion was injected with 10^7 PFUs of CAN-3110. Postmortem analyses were carried out 252 days after injection. Upper left panel shows MRI scan before post-mortem brain harvesting with necrotic injected occipital lesion, shown in the grossly necrotic lesion (middle upper panel), confirmed by histologic HE stain (right upper panel). The CAN-3110 non-injected temporal lobe postmortem gross section (lower right panel) exhibited oHSV positivity (middle lower panel) and dense infiltrates of CD8+ T cells. **Extended Data Figure 8** shows that

this oHSV positive focus was CAN-3110 and not reactivated latent wild-type HSV1 from this subject who otherwise was HSV seronegative throughout the trial. **(C)** HSV1 pathology staining in tumor tissue from rGBM/rHGG patients following CAN-3110 treatment relative to HSV1 serological status. **(D)** Same data as in panel C, but with patients that were initially seropositive grouped with patients that seroconverted following CAN-3110; with focal/weak pathology staining grouped with negative staining; and with multifocal staining grouped with positive staining. P value is from two-sided Fisher's exact test.

In the discussion, we added:

2- CAN-3110 persisted in injected tumors, with almost half of assayed rHGGs still positive even months after a single timepoint injection, but persistence significantly associated with negative HSV1 serology

And further in the discussion, we added:

We speculate that pre-existent HSV1 specific immunity enabled a more potent immune response possibly by innate responses and/or by rapid expansion of anti-HSV1 memory T cells in subjects challenged intratumorally with CAN-3110, whereas lack of anti-HSV1 immunity significantly reduced the ability to mount a rapid and effective immune response against the infected tumor (**Figure 7**).

And also:

The significant association of HSV1 seropositivity with absence of CAN-3110 antigen and transcripts in post-injection tumors suggests that an initial humoral and probably adaptive antiviral immune response led to an improved antitumor response based on the survival data and on the finding that there were still increased CD8+ and CD4+ T cells and increased immunologic transcriptional programs in tumors in spite of absent CAN-3110 in the longer surviving subjects (**Figure 7**).

We have added a new Figure 7 with corresponding legend to visualize the model we propose:

Figure 7 – A model for CAN-3110 action as a function of HSV1 serology. In HSV1 seropositive subjects, CAN-3110 elicits an initial augmented anti-HSV1 innate and T cell mediated response (presumably by expansion and differentiation of memory into effector anti-HSV1 T cells) to clear the injected oHSV from tumors. This “bystander” T cell effect possibly mediates an effective antitumor effect by direct inflammation in the tumor and/or by stimulating “antigen spreading” to also elicit T cell recognition of tumor antigens. In HSV1 seronegative subjects, the absence of a rapid anti-HSV1 innate and T cell response leads to CAN-3110 replicative persistence with tumor growth overcoming viral-induced cytotoxicity and delayed immune activity against tumor antigens.

-Figure 4: Is it possible to extract any information on the phenotype of T cells, eg are the T cells with enriched clones exhausted? memory? effector? naive?

Answer: We have re-analyzed the data from old **Figure 5**, which is now shown in revised **Figures 6** and **Extended Data Figures 11-12**. The re-analyses were performed in the context of positive HSV1 serology, showing that CAN-3110 induced a persistent change in injected rGBMs TME with significant changes in T cells with effector signatures and with changes in myeloid transcripts to M1. The results were thus changed to state:

“Tumor immune signatures correlate with survival. The observation that there was increased frequency of T cells in rGBMs that correlated with survival after CAN-3110 also prompted us to query RNA transcriptomic signatures in paired pre- and post-treatment tumor samples from 14 IDHwt rGBMs (13 patients, 14 interventions), for whom we had paired frozen tumors to maximize isolation of high-quality RNA. Amongst all subjects (both HSV1 seropositive and seronegative), there was a significant correlation between survival after CAN-3110 treatment and pre CAN-3110 transcriptomic signatures related to T reg trafficking, T cells, MHCI and II, effector cells, T cell trafficking and cancer associated fibroblasts (CAFs) (**Extended Data Figure 11A**). In the same paired tumors, there was a significant correlation between survival and post CAN-3110 transcriptomic signatures related to antitumor cytokines, M1, MDSC, and T cell signatures (**Extended Data Figure 11B**). Interestingly, while correlations between survival and pre-treatment immune signatures were weaker when analyzed using only HSV1 serologically positive subjects (reflecting the decreased sample size) (**Extended Data Figure 12A**), correlations between survival and post-treatment signatures were notably stronger when analyzed using HSV1 serologically positive subjects rather than all subjects (**Extended Data Figure 12B, Figure 6A-B**). In HSV1 seropositive patients, post-treatment signatures for antitumor cytokines, M1, checkpoint inhibition, MDSC, MDSC trafficking, T cells, T cell trafficking, coactivation molecules, macrophages, macrophage DC trafficking, effector cells, Th1, and MHCI were all significantly correlated with post CAN-3110 survival (**Extended Data Figure 12B, Figure 6A&C**). Furthermore, when focusing on HSV1 seropositive patients, many immune gene signatures associated with immune activation (i.e. antitumor cytokines, T cell traffic, coactivation molecules, etc.) were much more strongly correlated with survival post-treatment than they were pre-treatment (**Figure 6C**). When considered together with other data from this study (**Figure 6D**), these results demonstrate that CAN-3110 instigates a highly inflammatory and immunologically activated tumor microenvironment in HSV1 serologically positive patients which persists beyond detectable HSV1 antigen and is significantly correlated with post-treatment survival in a way that is not true of the pre-treatment tumor immune state.”

Revised **Figure 6** (old **Figure 5**) and corresponding legend now are revised to:

“Figure 6 - Survival correlation between immune transcript signature programs in HSV1-seronegative and seropositive subjects. Thirteen paired IDHwt rGBMs with good quality RNA were analyzed by bulk RNA transcriptomics. Transcriptomic signatures for different biologic programs were estimated for each sample, and these signatures were assessed for correlation with post CAN-3110 survival using either all patients or only HSV1 seropositive patients. (A) Example of two immune signatures (Anti-tumor cytokine and T-cell signatures) which are more strongly correlated with post CAN3110 survival when analyzed using only HSV1 seropositive patients rather than with all patients. (B) The change in Pearson’s correlation p (left) and r (right) values

when analyses were performed using all patients (red points) or only HSV1 seropositive patients (teal points). Only gene signatures which reached $p \leq 0.05$ (dashed red line) in either analysis are plotted. (C) Change in Pearson's correlation p (left) and r (right) values for pre-treatment (red points) and post-treatment (teal points) samples from HSV1 seropositive patients. This panel includes all analyzed RNAseq gene signatures. The dashed red line indicates $p = 0.05$. For C-D, HSV1 serology remained unchanged following CAN3110 for all patients, and one patient (045) was omitted from the analysis due to early non-GBM mortality. (D) Combined data for all patients in the study, including post CAN-3110 survival, HSV1 serology, HSV1 tumor pathology, T cell fraction changes based on TCR β DNA sequencing, initial tumor volumes, post-treatment tumor volumetric growth rates, and bulk RNAseq based antitumor cytokines signature scores. Grey boxes indicate missing data.”

In addition, we have also performed single cell RNA sequencing on some tumors, showing that T cells in injected tumors express an effector more than a memory or exhausted signature. Because these results are quite exhaustive in terms of showing the T cell phenotype, we plan to include these results in a separate publication.

-Supplemental fig 13 shows the main checkpoint inhibitors such as PD1 and CTLA4, however other markers, which are considered hallmarks of intratumoral tumor-reactive CD8 T cells are missing, such as Tim-3, CD38, TCF-1 CD103, CD39 (among others).

Answer: We thank the reviewer for this excellent comment. While post-treatment expression of many of these genes was not significantly correlated with post-treatment survival (see figure below), this prompted us to re-analyze gene signatures of immune activation. This analysis revealed that many post-treatment signatures of immune activation are correlated with post-treatment survival, particularly in HSV1 seropositive patients. This new data has been added in new **Figures 6** (see previous answer and excerpt from new Figure 6 showing TIM3, CD39, CD103, TCF1 below) and **Extended Data Figures 11-12**. The results and discussion were revised as in previous answer.

-For the assessment of TCR diversity it could also be of interest to determine CDR3 length of TILs between LS and SS

Answer: We thank the reviewer for this suggestion. While we did not find any statistically significant differences between CDR3 length distributions in TILs for LS and SS (see plot below for reviewer).

IDHwt rGBMs (n = 21)

However, this prompted us to take a new look at V(D)J gene usage as an alternative method of quantifying TCR diversity changes in TILs. This resulted in the identification of a correlation between post-treatment TCRBV09-01*01 usage and survival in HSV1 seropositive patients (now added as new **figure panel 5D**). The following text was added in the Results:

“When assessing V(D)J gene usage, we also identified a correlation between post-treatment TCRBV09-01*01 usage and survival in HSV1 seropositive patients (Figure 5D, Pearson’s $r = 0.00034$, FDR = 0.017).”

Figure 5. (D) Correlation between post-treatment TCRBV09-01*01 usage and post-treatment survival in IDHwt rGBM patients (excluding 045 due to non-GBM mortality). Pearson’s correlation coefficients r and p values are shown above the plot when calculated using all patients or using only HSV1 seropositive + seroconverted patients. FDR correction was performed using all V genes with a median usage > 0 in both pre- and post-treatment samples from these patients.

Minor comments:

-Why not also show HSV preexisting antibodies in IDH mutant patients (n=9 in total) to see if same trend is found?

Answer: We have added an additional figure (Extended Data new Figure 5B—shown below) to show this. We have also added the following to the main text.

“In a survival analysis, HSV1 seropositive patients lived a median of 14.2 months (95% CI: 9.5 – 16.7) vs. only 7.8 months (95% CI: 4.9 – Inf) for seronegative patients (p = 0.008, likelihood ratio test, **Figure 1C**). Contrastingly, HSV2 serology was not associated with survival (p = 0.9, likelihood ratio test, **Figure 1D**). Similarly, the trend towards longer survival for HSV1 seropositive patients was observed in the IDH mutant rAA patients (**Extended Data Figure 5B**), though the sample size for this group is too small to draw definitive conclusions.”

-Fig 2C peri-necrotic core staining of CD8 and CD4 would benefit from insertion of selected area enlargements at higher magnification to show specific staining.

Answer: Thank you for this suggestion. Areas of enlargement have been added to this panel, which is now revised Figure 2C.

-Fig 2D Mention type of cell (CD4, CD8, CD20) above the graphs for easier reading

Answer: These labels have been added in what is now revised as Figure 2D.

-Fig 2E what about pre peri-necrotic?

Answer: The pre- samples always consist of stereotactic biopsies to confirm presence of glioma before injection of CAN-3110. Therefore, the quantity of tissue precludes extensive analyses of areas of necroses. In addition, the biopsies are routinely carried out into areas of gadolinium-enhancement rather than areas of necrosis to maximize the possibility of finding tumor cells.

-Present CD4, 8, and 20 in same order each figure for clarity

Answer: We have rearranged the panels in now revised Figure 2 to keep the same order throughout for clarity.

- Add to X axis title of 3A and 3B 'in tumor' and 'in PBMCs' for easier graph reading

Answer: We have now added "Tumor" and "PBMC" labels to the axes of all plots in this figure, which is now revised Figure 4.

-Extended data file page 12, replace brain tissue collection by brain tumor tissue collection

Answer: Thank you for this correction. The change has been made (now in **Supplementary Methods**).

Referee #4 (Remarks to the Author):

This manuscript describes an innovative and a well conducted first in human phase I trial of CAN-3110, an oncolytic herpes virus strain that expresses the viral neurovirulence ICP34.5 gene under the transcriptional control of the Nestin promoter in patients with recurrent GBM. In addition, the manuscript presents a before and after treatment comprehensive correlative analysis of tumor and peripheral blood including serologic, immunohistochemistry as well as RNA and T cell receptor DNA sequence analysis. Overall, the findings provide novel insights into the mechanism of action of the CAN-3110 that have broad translational implications for the oncolytic virotherapy field.

Comments to Authors:

- The investigators have obtained post treatment tumor tissue at different time points. It would be important to know what other antitumor treatments the patients have received prior to the surgical sampling being performed since this could have impacted the correlative analysis results.

Answer: These can be found in **Extended Data Figures 3C-D** with swimmers plots as well as new **Supplementary Table 1**. To make it easier to read, we have added to the text in the results (new text underlined):

“Progression-free survival times for the entire cohort and divided by the 3 rHGG diagnostic groups are shown in **Extended Data Figures 3A and B**, respectively, and treated subjects’ clinical course is illustrated in **Extended Data Figures 3C-D**. Note that in the swimmers’ plots, the timepoint of post-injection tumor resection is illustrated by a colored triangle, with most additional antitumor therapies administered after resection. Full patient treatment histories have been included in **Supplementary Table 1**. Examples of significant clinical and radiographic responses are illustrated in **Extended Data Figure 4**, including a response in a multifocal/ multicentric rGBM.”

- For patients in whom surgical sampling on more than one occasions was performed, it would be desirable to see individual patient trends over time regarding T cell infiltrates, RNA and TCR DNA sequence analysis.

Answer: Unfortunately, RNAseq signatures and TCR DNA sequencing were only performed at a single post-treatment timepoint for each patient—precluding observation of individual patient trends over time in these datasets. However, we have added new **Extended Data Figures 7A-B** to show changes in IHC quantifications of CD8+, CD4+, and CD20+ cells in tumor samples across time for individual patients. The following paragraph has been added to the Results section to discuss this data:

Although longitudinal analyses showed perhaps a non-significant trend in reduction of CD8+ T cell numbers over time in HSV1 seronegative patients (Kruskal-Wallis $p = 0.12$, **Extended Data Figures 7A-B**), there was no time-dependent trend in reduction in CD8+ T cell numbers in HSV1 seropositive patients ($p = 0.49$), possibly suggesting that the tumor immune response induced by CAN-3110 may be durable.

Wherever possible, we have also shown how tissue collection timepoint affected metrics when looked at across all patients in aggregate. This data is included in new **Extended Data Figures 7C-E** for IHC quantifications used in new **Figure 2F**; in new **Extended Data Figure 9A** for tumor T cell fraction/diversity metrics from TCR sequencing data; and in new **Extended Data Figure 11C** for RNAseq gene signatures.

- The investigators comment that in 12/29 tumors HSV antigen was detected by IHC even several months after CAN-3110 injection. PCR data confirming the presence of CAN-3110 specific viral RNA was only presented for

one patient, however. Was PCR performed in all patients with HSV positivity in IHC and what were the respective findings?

Answer: we have not performed PCR in all subjects where HSV antigen was detected in tumor because the large majority of these were HSV seronegative (see new **Figure 3C-D**) and thus the detected viral antigen was very unlikely to be an HSV other than injected CAN-3110. In addition, most of these HSV immunopositive areas anatomically correlated closely to the areas of CAN-3110 injection

- Increased TCR β diversity was noted. It would be important to know if this reflects antitumor or antiviral immune response.

Answer: We agree with the reviewer. This is a very important area that is subject to continuous future research by both us and other labs. It requires extensive experimentation including TCR reconstruction strategies, TCR expression and library screening of epitopes recognized by these reconstructed TCRs. We would argue that such lengthy experimentation and absence of this data from the current paper does not detract from the findings and validating the hypotheses of the current paper.

- What is the definition of long-term and short-term survivors for the data presented in Supplementary Figure 13C and D? It is possible that other treatments that these patients have received after progression on study could have impacted these results that they cannot solely be attributed to the viral treatment.

Answer: In the original figure legend we had defined LS as subjects who survived longer than one year. In the new revised version, these results were re-analyzed and are now presented as new **Figure 6** and **Extended Data Figures 11-12**. The treatments received by subjects at suspected progression after CAN-3110 (depicted in the swimmers' plots of **Extended data Figures 3C-D** and also listed in new **Supplementary Table 1**) were multiple, heterogenous, and none shown to affect rGBM outcome, with CAN-3110 being the only common treatment. Therefore, although we cannot exclude that these other treatments could have affected the length of survival, we argue that the above considerations make it unlikely.

- Page 8: “We found that post-treatment survival was significantly longer for patients who were serologically HSV+ pre- or post-treatment with CAN-3110.” Please include median OS and range for the two groups in addition to p-values.

Answer: We thank the reviewer for pointing out that this section of the results lacked important information and was potentially confusing to read. We have changed the relevant section to the following:

“Surprisingly, we found that both pre- and post-treatment HSV1 seropositivity were associated with significantly longer post-treatment survival ($p = 0.011$ and $p = 0.016$, respectively) (**Extended Data Figure 5A**). In a survival analysis, HSV1 seropositive patients lived a median of 14.2 months (95% CI: 9.5 – 16.7) vs. only 7.8 months (95% CI: 4.9 – Inf) for seronegative patients ($p = 0.008$, likelihood ratio test, **Figure 1C**). In contrast, HSV2 serology was not associated with survival ($p = 0.9$, likelihood ratio test, **Figure 1D**).”

Reviewer Reports on the First Revision:

Referees' comments:

Referee #1 (Remarks to the Author):

The revised manuscript by Ling et al is original and significant and provides highly relevant data for the cancer field in general, and oncolytic virus field in particular.

Data and methodologies are robust, of high quality and clearly presented.

Previous comments have been (extensively) addressed.

Referee #4 (Remarks to the Author):

I had the opportunity to review the revised manuscript. The additional analysis presented strengthens the manuscript and provides additional mechanistic insights.

I have a few additional comments:

The data that the authors present suggest that the clinical benefit of CAN-3110 is primarily limited to HSV-1 seropositive patients.

The authors should discuss further correlation of HSV-1 seropositivity with efficacy in the context of brain tumor trials with other HSV strains; is this a class effect or only applicable to the CAN-3110 strain? It would be also helpful to provide a recommendation on how/if these findings should impact eligibility in future trials and applicability of the approach.

Additional information should be provided regarding patient 19, including imaging documentation of TME remodeling (since per the authors' response, these have only been presented in meeting forums). Also, this data create additional questions regarding the mechanism of action and if viral replication is necessary.

Author Rebuttals to First Revision:

Responses to Referees' comments:

Referee #1 (Remarks to the Author):

The revised manuscript by Ling et al is original and significant and provides highly relevant data for the cancer field in general, and oncolytic virus field in particular.

Data and methodologies are robust, of high quality and clearly presented.

Previous comments have been (extensively) addressed.

Response: We thank the referee

Referee #4 (Remarks to the Author):

I had the opportunity to review the revised manuscript. The additional analysis presented strengthens the manuscript and provides additional mechanistic insights.

I have a few additional comments:

The data that the authors present suggest that the clinical benefit of CAN-3110 is primarily limited to HSV-1 seropositive patients.

The authors should discuss further correlation of HSV-1 seropositivity with efficacy in the context of brain tumor trials with other HSV strains; is this a class effect or only applicable to the CAN-3110 strain? It would be also helpful to provide a recommendation on how/if these findings should impact eligibility in future trials and applicability of the approach.

Answer: We thank the reviewer. To address this excellent point. In the discussion on page 15 we state:

“The finding that positive HSV1 serology before or after CAN-3110 was a highly significant independent predictor of response was unexpected, based on previously reported trials of other oHSVs^{16,17,19,31}. A recent study showed no correlation between HSV1 serology in humans with GBM and survival⁴¹. We speculate that this discovery may be OV specific, based on the capacity of each OV to replicate, persist, and stimulate an innate and adaptive immune response. It may also be a factor related to sample size, at least for the brain tumor trials, since our trial had more subjects. It should be noted that the 20 subjects with rGBM IDHwt that were HSV1 serologically positive before CAN-3110 had a mOS of 14.2 months (95% CI: 9.5 – 16.7, see **Figure 1C**), which is higher than the historical mOS of 6-9 months. Further prospective validation of this discovery in the next phase of planned trials will determine if HSV1 serology can be used as a selection criterion for subject likelihood of response.”

Additional information should be provided regarding patient 19, including imaging documentation of TME remodeling (since per the authors' response, these have only been presented in meeting forums). Also, this data create additional questions regarding the mechanism of action and if viral replication is necessary.

(Redacted)

In terms of the comments related to viral replication, we believe it is needed in order to increase the viral burden and the immune viral cytotoxic effect, which would enhance the antiviral and antitumor T cell response. Lack of or poor replication may limit this mechanism. Even without nestin expression, CAN-3110 would still replicate, albeit in a much more attenuated fashion, like other ICP34.5 null oHSV1s.